# Antimicrobial and anti-biofilm activity of a thiazolidinone derivative against *Staphylococcus aureus in vitro* and *in vivo*

Rui Zhao,[1] Bingyu Du,[1] Yue Luo,[2] Fen Xue,[1] Huanhuan Wang,[3] Di Qu,[4] Shiqing Han,[2] Simon Heilbronner,[5] Yanfeng Zhao[1]

**ABSTRACT** *Staphylococcus aureus* (*S. aureus*) causes many infections with significant morbidity and mortality. *S. aureus* can form biofilms, which can cause biofilm-associated diseases and increase resistance to many conventional antibiotics, resulting in chronic infection. It is critical to develop novel antibiotics against staphylococcal infections, particularly those that can kill cells embedded in biofilms. This study aimed to investigate the bacteriocidal and anti-biofilm activities of thiazolidinone derivative (TD-H2-A) against *S. aureus*. A total of 40 non-duplicate strains were collected, and the minimum inhibitory concentrations (MICs) of TD-H2-A were determined. The effect of TD-H2-A on established *S. aureus* mature biofilms was examined using a confocal laser scanning microscope (CLSM). The antibacterial effects of the compound on planktonic bacteria and bacteria in mature biofilms were investigated. Other characteristics, such as cytotoxicity and hemolytic activity, were researched. A mouse skin infection model was used, and a routine hematoxylin and eosin (H&E) staining was used for histological examination. The MIC values of TD-H2-A against the different *S. aureus* strains were 6.3–25.0 µg/mL. The 5 × MIC TD-H2-A killed almost all planktonic *S. aureus* USA300. The derivative was found to have strong bacteriocidal activity against cells in mature biofilms meanwhile having low cytotoxicity and hemolytic activity against Vero cells and human erythrocytes. TD-H2-A had a good bacteriocidal effect on *S. aureus* SA113-infected mice. In conclusion, TD-H2-A demonstrated good bacteriocidal and anti-biofilm activities against *S. aureus*, paving the way for the development of novel agents to combat biofilm infections and multidrug-resistant staphylococcal infections.

**IMPORTANCE** *Staphylococcus aureus*, a notorious pathogen, can form a stubborn biofilm and develop drug resistance. It is crucial to develop new anti-infective therapies against biofilm-associated infections. The manuscript describes the new antibiotic to effectively combat multidrug-resistant and biofilm-associated diseases.

**KEYWORDS** *Staphylococcus aureus*, bacteriocidal, anti-biofilm activity, WalK

S taphylococcus aureus (*S. aureus*) is a major commensal bacterium and a human pathogen that causes a hard-to-estimate number of uncomplicated skin infections and hundreds of thousands to millions of severe invasive infections per year (1–3). Antibiotic resistance is a serious global concern, and new treatments are urgently needed (4). Along with the widespread use of different antibiotic classes, the emergence of methicillin-resistant *S. aureus* (MRSA) has posed a significant therapeutic challenge. Vancomycin, a glycopeptide antibiotic used to treat Gram-positive bacterial infections, has been widely used as first-line therapy for MRSA, increasing the number of resistant strains, including vancomycin-intermediate (minimum inhibitory concentration (MIC): 4–8 µg/mL), vancomycin-resistant *S. aureus* (MIC ≥16 µg/mL), and vancomycin-resistant *Enterococcus* (VRE). The emergence of these resistant strains has prompted a more diligent approach to monitoring and restricting vancomycin use (5–7).

Address correspondence to Yanfeng Zhao, zhaoyanfeng@njmu.edu.cn, or Shiqing Han, hanshiqing@njtech.edu.cn.

Rui Zhao and Bingyu Du contributed equally to this article. Author order was determined by drawing straws.

The authors declare no conflict of interest.

In addition to specific antibiotic resistance, nonspecific antibiotic resistance caused by biofilm formation plays a role in many *S. aureus* infections (8). Bacteria embedded in a biofilm exist in a low metabolic state or a stationary growth phase; the biofilm matrix consists of glycopolymers, proteins, and extracellular DNA, enabling bacteria to resist the host immune response and escape antibiotic killing. In fact, bacteria in a biofilm can be 1,000 times more resistant to antimicrobial agents than their planktonic counterparts (9). Nowadays, biofilm infections are not only a problem in the healthcare sector but also a major global challenge (10). Thus, the identification of novel therapeutic targets to fight biofilm-related infections is one of the primary problems in the field of antibiotic therapy (11).

Two-component systems (TCSs), which consist of a histidine kinase (HK) sensor and a response regulator, are important for bacteria to quickly sense and respond to environmental signals (12). Because no HK is found in mammals, including humans, TCSs could be identified as therapeutic targets. The WalK/WalR (YycG/YycF) TCS, discovered in *Bacillus subtilis*, is highly conserved and specific to low G + C Gram-positive bacteria such as *S. aureus* and *Staphylococcus epidermidis* (*S. epidermidis*). WalK is a sensor HK, and WalR is the cognate response regulator. In *S. aureus*, the WalKR TCS is an essential regulatory system for cell wall metabolism, including autolysis, biofilm formation, capsule synthesis, and virulence. It is highly conserved (13), making it an attractive drug target for the development of novel anti-bacterial drugs.

In a previous study (14), we used the cytoplasmic HATPase-c domain of histidine kinase YycG (WalK) in *S. epidermidis* as an antimicrobial target. We designed, synthesized, and screened three thiazolidinone compounds (Compounds 2, 5, and 7) with the best effect on bacteriocidal and biofilm-killing activities, and then the functional groups of Compound 2({2-{4-{3-(2-ethylphenyl)−2-[(2-ethylphenyl)imino]−4-oxothiazolidin-5-yli-dene}methyl}−2-methoxyphenoxy}acetic acid) were modified while the thiazolidine core structure remained intact (15, 16). A series of derivatives were designed and synthesized. Compounds with strong bacteriocidal and anti-biofilm activities were screened from 95 derivatives of Compound 2, and their bacteriocidal activity was evaluated (14, 17, 18). Our team previously screened a new derivative, TD-H2-A (The structural formula is shown in Fig. 1), and 40 clinical and laboratory strains were collected and used to evaluate the anti-bacterial activities of TD-H2-A against clinical *staphylococcal* isolates under planktonic and biofilm growth conditions. We also described a mouse model of skin and soft tissue infection that we used to conduct a series of *in vivo* animal experiments.

## RESULTS

### MIC definition

In this experiment, anti-bacterial activities against 40 strains that belong to different species and genera, seven of which were drug-resistant strains, were assessed. The MIC values of TD-H2-A against these clinical isolates were determined using the broth microdilution method and ranged from 0.06 to 100 µg/mL (Table 1). The newly synthesized derivative TD-H2-A demonstrated obvious antimicrobial activity against *S. aureus*, *Staphylococcus warneri*, *Staphylococcus lungdunisis*, *S. epidermidis*, *Streptococcus pneumoniae*, *Streptococcus constellatus*, *Enterococcus faecalis*, and *Bacillus subtilis*. In addition, TD-H2-A had potent activities against multidrug-resistant strains, including vancomycin or methicillin-resistant strains, such as MRSA, glycopeptide intermediate-resistant *S. aureus* (GISA), and VRE. The anti-staphylococcal activities of TD-H2-A were greater than the activities against *Corynebacterium striatum* and *Listeria monocytogenes* strains. At concentrations of 25 µg/mL or less, TD-H2-A inhibited the growth of all staphylococcal isolates.

**FIG 1** The structure of TD-H2-A.

## Cytotoxicity and hemolytic analysis

After treatment with different concentrations of TD-H2-A, the survival rates of Vero cells showed a decreasing trend and showed a dose-dependent relationship. But even at a concentration of 100 µg/mL (exceeding its MIC against staphylococci 4- to 31-fold) the survival rate of Vero cells was 73.26% (Fig. 2A). The MTT assay revealed that the TD-H2-A had no obvious inhibitory effect on the growth of Vero cells. The hemolytic effect of the new synthetic derivatives was assessed on healthy human erythrocytes. Concentration below the MIC did not have an obvious hemolytic effect on the hemolysis ratio of human red blood cells, which was less than 0.67%. At a higher concentration of 25 µg/mL, the hemolysis ratio of the TD-H2-A derivative was less than 1.42% (Fig. S1A). The results showed that the TD-H2-A had no obvious hemolytic effect on the red blood cells of healthy people at the concentration used. As a result, TD-H2-A caused minimal cytotoxicity to Vero cells and minimal hemolytic activity against human erythrocytes.

## Resistance and membrane permeability

To determine whether TD-H2-A-resistant colonies can be generated in *S. aureus* HG001, the cells were serially passaged in a TSB medium with daily increasing TD-H2-A concentrations for 25 days. The maximum TD-H2-A concentration at which *S. aureus* could grow was 4 × MIC (25 µg/mL; Fig. 2B). By contrast, after 10 days of treatment with the gyrase inhibitor ofloxacin, *S. aureus* cells acquired resistance to 128 × MIC (32 µg/mL). The TD-H2-A MIC increased fourfold in these strains after serial passage, and a fourfold increase in MIC indicates reduced susceptibility. Resistance development results revealed that *S. aureus* had no resistance to TD-H2-A after successive treatment of *S. aureus* HG001 over 25 days with TD-H2-A. After 25 days of continuous culture, the derivative could inhibit bacterial growth at 4× MIC, and the bacteriocidal effect was superior to that of ofloxacin. By comparing the whole-genome sequencing of HG001 treated with 4× MIC TD-H2-A on the 25th day and not treated with TD-H2-A, mutations have occurred in six genes, including *BSR30_RS01405*, *rrf*, *mnhD1*, *BSR30_RS05085*, *BSR30_RS07385*, and *BSR30_RS12160*, and the sequencing comparison results are shown in Table S1.

As we can see in Fig. 2C, the compound affected the membrane permeability at different time points, and the difference became more obvious as the concentration increased and time went on. The membrane permeability experimental results (Fig. 2C)

**TABLE 1** Compound spectrum of activity[a]

| Species and strain | Resistance | TD-H2-A MIC (µg/mL) |
|---|---|---|
| *Staphylococcus aureus* USA300JE2 | MRSA | 12.5 |
| *Staphylococcus aureus* Newman | | 6.3 |
| *Staphylococcus aureus* M15-5 | | 12.5 |
| *Staphylococcus aureus* SA113 | | 25.0 |
| *Staphylococcus aureus* SH1000 | | 25.0 |
| *Staphylococcus aureus* M11-28 | | 12.5 |
| *Staphylococcus aureus* M13-14 | | 12.5 |
| *Staphylococcus aureus* ST45 CD140657 | MRSA | 25.0 |
| *Staphylococcus aureus* Mu50 | GISA | 25.0 |
| *Staphylococcus aureus* Nr.1764 NRS184 | | 25.0 |
| *Staphylococcus aureus* Nr.1767 NRS229 | | 12.5 |
| *Staphylococcus aureus* Nr.1763 NRS71 | MRSA | 12.5 |
| *Staphylococcus aureus* ST398 82,086 WT | MRSA | 12.5 |
| *Staphylococcus aureus* ST398 105 | | 12.5 |
| *Staphylococcus aureus* ST398 124 | | 6.3 |
| *Staphylococcus aureus* ST398 103 | | 25.0 |
| *Staphylococcus aureus* RH45 | | 25.0 |
| *Staphylococcus aureus* RH47 | MRSA | 12.5 |
| *Staphylococcus aureus* HG001 | | 6.3 |
| *Staphylococcus aureus* 2109180 | | 25.0 |
| *Staphylococcus warneri* 35 | | 25.0 |
| *Staphylococcus lugdunensis* IVK28 | | 25.0 |
| *Staphylococcus epidermidis* 39 | | 6.3 |
| *Staphylococcus epidermidis* 2184498 | | 25.0 |
| *Staphylococcus epidermidis* 2130437 | | 25.0 |
| *Streptococcus pneumoniae* 2132073 | | <=0.06 |
| *Streptococcus pneumoniae* 2131813 | | <=0.06 |
| *Streptococcus constellatus* 2131630 | | 6.3 |
| *Enterococcus faecalis* VRE366 | VRE | 12.5 |
| *Bacillus subtilis* 508 | | 6.3 |
| *Bacillus subtilis* 509 | | 6.3 |
| *Corynebacterium striatum* 2129344 | | 50 |
| *Corynebacterium striatum* 2129003 | | 50 |
| *Corynebacterium striatum* 2129005 | | 50 |
| *Listeria monocytogenes* 2131345 | | 100 |
| *Listeria monocytogenes* 2184650 | | 100 |
| *Listeria monocytogenes* 203003 | | 100 |
| *Klebsiella pneumoniae* | | >200 |
| *Pseudomonas aeruginosa* | | >200 |
| *Escherichia coli* | | >200 |

[a]The MIC was determined by broth microdilution. MRSA, methicillin-resistant *S. aureus*; GISA, glycopeptide intermediate-resistant *S. aureus*; VRE, vancomycin-resistant *Enterococcus*.

revealed that 1× MICs of TD-H2-A had no significant effect on the membrane permeability of *S. aureus* HG001 at time point 60 minutes ($P = 0.0508$), while 4× or 8× MICs of TD-H2-A increased permeability significantly at different time points ($P < 0.01$), and nisin significantly increased permeability at 5× MIC. TD-H2-A increased permeability obviously after 180 min from Fig. 2C, whereas nisin showed increased permeability quickly already at 30 min.

## Bacteriocidal effect of thiazolidione derivative

Since the MIC values of TD-H2-A and vancomycin on MRSA (USA300) have been established, we compared TD-H2-A's efficacy in killing mid-exponential *S. aureus* USA300

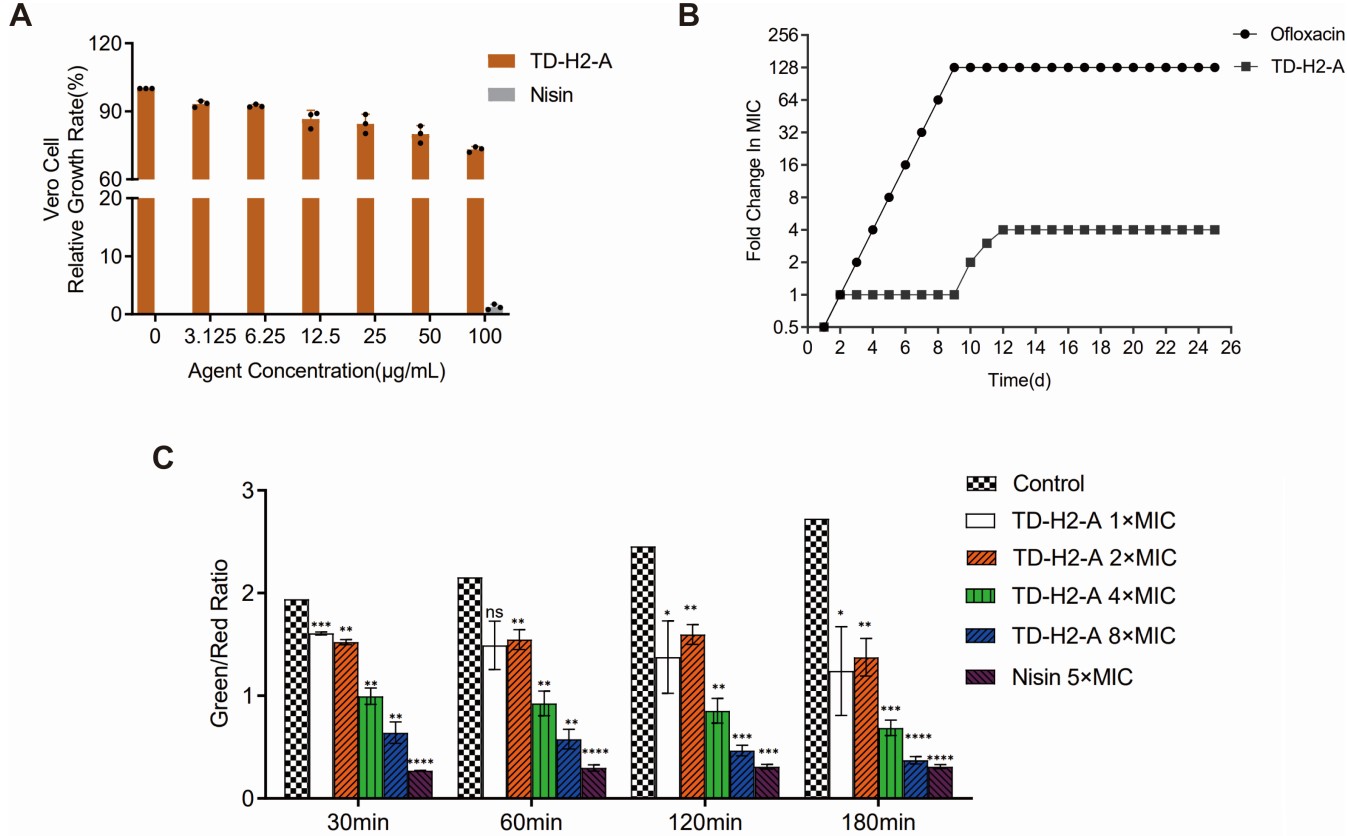

FIG 2 (A) Cytotoxicity of TD-H2-A toward Vero cells. (B) Sequential passaging broth assay for the development of spontaneous antibiotic resistance. The y-axis and course of the curve represent the antibiotic concentration at which the cells could grow during passaging. The graph shown in the figure is from a single experiment that has three replicates. (C) Membrane permeability was monitored by measuring the fluorescence of SYTO9, a green fluorescent nucleic acid stain, and propidium iodide, a red fluorescent nucleic acid stain. 1% DMSO was used as a negative control, while nisin was used as a positive control. The excitation wavelength of 485 nm and emission wavelength of 620/530 nm were for red and green fluorescence, respectively. All assays were performed with three biologically independent experiments, and the mean ±SD is shown. Statistical differences were analyzed by $t$-test and $F$-test (***$P$ < 0.001, **** indicates $P$ < 0.0001, and "ns" indicates $P$ > 0.05).

and biofilm cells of *S. aureus* SA113 with vancomycin. First, a time-killing assay was performed with TD-H2-A and vancomycin at 2× MIC, 4× MIC, 5× MIC, 6× MIC, 8× MIC, 10× MIC concentrations. All groups began with $3.1 \times 10^6$ CFU/mL viable bacterial cells. The bactericidal effect of TD-H2-A was found to be effective, with a marked reduction of living planktonic bacteria of *S. aureus* USA300 at 5× MIC. Notably, TD-H2-A quickly killed bacteria and eradicated the mid-exponential population of *S. aureus* and in the first 4 hours the living strains reduced by 99.9%, and the number of viable bacteria almost reached the detection limit at 5× MIC in 24 hours, with few living bacteria detected. The strains in the control group had been steadily growing (Fig. 3A through F).

This viability assay of bacteria encased in biofilms revealed that the compound TD-H2-A outperformed vancomycin in killing established *S. aureus* biofilm cells. As shown in Fig. 4A, TD-H2-A reduced the viability of *S. aureus* SA113 biofilm cells in a concentration-dependent manner. The results showed that there was no significant difference between vancomycin (20 µg/mL, 10 × MIC) and 1% DMSO (the control group). The number of bacteria in mature biofilms was significantly reduced compared to the control group (Treatment with the small molecule compound TD-H2-A of 1×, 5×, and 10× MIC). The killing potency of compound TD-H2-A increased, with a reduction in viable bacteria numbers from $10^7$ to $10^4$, as TD-H2-A concentrations increased from 1 × to 10 × MICs (Fig. 4A).

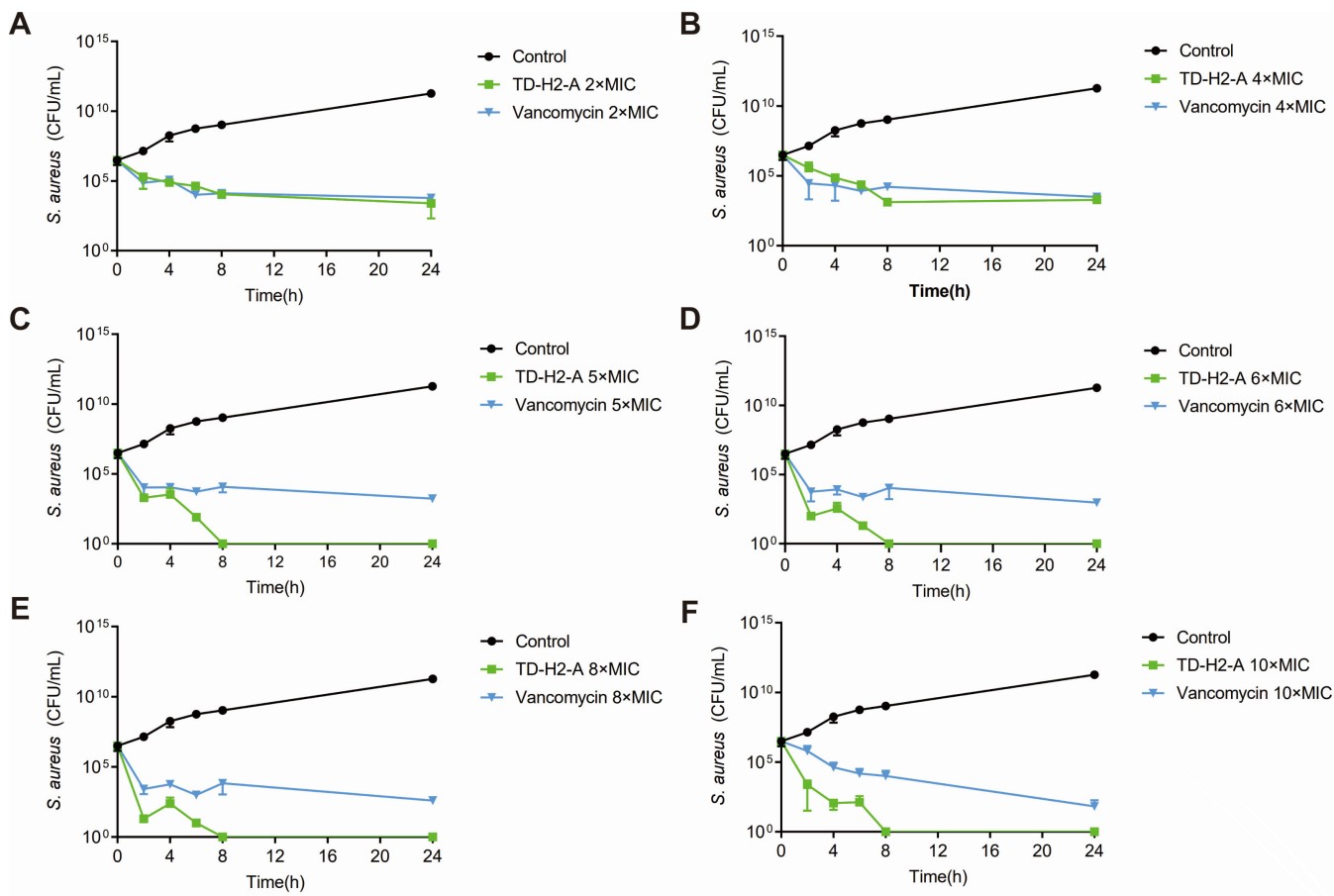

**FIG 3** Bacteriocidal activity of TD-H2-A. Time-killing curve of TD-H2-A and vancomycin against *S. aureus* USA300 in 2× MIC, 4× MIC, 5× MIC, 6× MIC, 8× MIC, and 10× MIC. Samples were taken at the indicated time points, and the number of living cells was determined by the CFU counts. The data presented was the average of three independent experiments (mean ± SD).

The effects of the derivative on cell viability in mature *S. aureus* biofilms were detected using a CLSM with Live/Dead staining. The control group results showed that almost all of the bacteria in the biofilm were viable (stained green by SYTO9), and vancomycin (20 µg/mL, 10 × MIC) had little effect on bacterial viability in biofilms, reducing living cells in mature biofilms by only 3.4%. The majority of bacteria in mature biofilms treated with TD-H2-A at 1 × MIC concentration were viable bacteria. However, the derivative in 5×, 10× MIC reduced the proportion of viable cells in mature biofilms to varying degrees. In the 10× TD-H2-A treatment group, more than 99.4% of the bacteria in the biofilm were stained red by PI, indicating that the majority of the bacteria were dead (Fig. 4B).

## Mouse infection model and pathological sections

To evaluate the effect of TD-H2-A *in vivo*, we used a mouse skin infection model in which the mice were injected with TD-H2-A, vancomycin, and 1% DMSO, respectively. We also performed hematoxylin and eosin (H&E) staining to assess the treatment effect. Comparing each treatment group (vancomycin group and TD-H2-A group) to the 1% DMSO control group, there was no significant difference between the vancomycin and control groups ($P > 0.05$), while the number of SA113 viable bacteria in the skin of mice of the TD-H2-A group was significantly reduced which had a fourfold reduction ($P = 0.0005$; Fig. 5A).

H&E staining revealed that *S. aureus* SA113 induced infection in mice skin tissues, whereas control mice that were infected with *S. aureus* but received only vehicle (1%

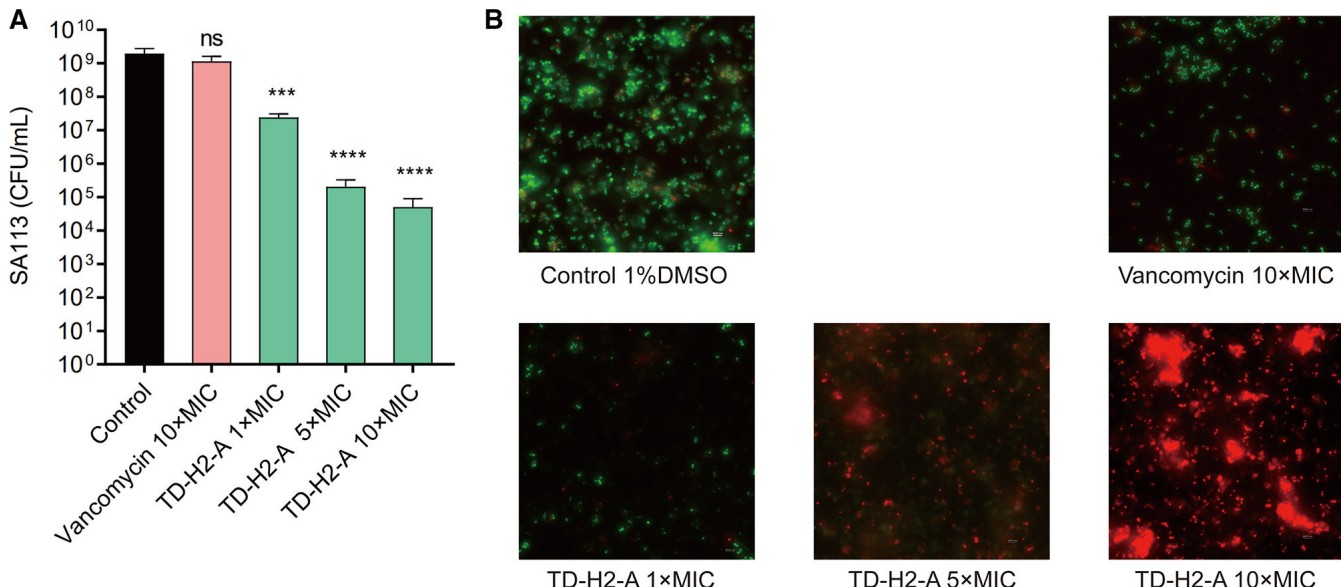

**FIG 4**  (A) Effect of TD-H2-A on 24-hour-old *S. aureus* SA113 biofilm cells. After treatment of a 24-hour-old *S. aureus* SA113 biofilm with TD-H2-A and vancomycin, the viability of the biofilm cells was determined by the CFU counts. The data are presented as the mean and standard deviation (SD) of three independent experiments. The data were analyzed using one-way ANOVA with multiple comparisons of treatments to control. Only significant differences are shown, with the green solid line representing TD-H2-A treatment and the pink solid line representing vancomycin treatment. *** indicates $P < 0.001$, **** indicates $P < 0.0001$, and "ns" indicates $P > 0.05$. (B) Confocal images of *S. aureus* biofilms following viability staining showing the effects of TD-H2-A on the mature biofilm of *S. aureus* SA113. The established biofilms of *S. aureus* SA113 stained with SYTO9 and propidium iodide (PI) were examined under a CLSM, Leica). Cells stained with green fluorescence (SYTO9) were viable, while those stained with red fluorescence (PI) were dead. The experiments were performed in triplicate, and representative images are shown.

DMSO) had obviously positive staining. On Day 4, representative histological images revealed that the skin epidermis was ruptured in the 1% DMSO control group, with inflammatory exudation on the surface and a large number of neutrophils infiltrated in the dermis and subcutaneous adipose tissues, with abscess formation and necrosis. In the vancomycin treatment group, there was still inflammatory exudation, inflammation in subcutaneous adipose tissue, and a large number of neutrophils infiltrated with abscess and necrotic tissue formation. In the TD-H2-A treatment group, the skin was intact, with small abscesses in the focal epidermis and no obvious neutrophil or lymphocyte infiltration in the dermis or subcutaneous adipose tissues (Fig. 5B).

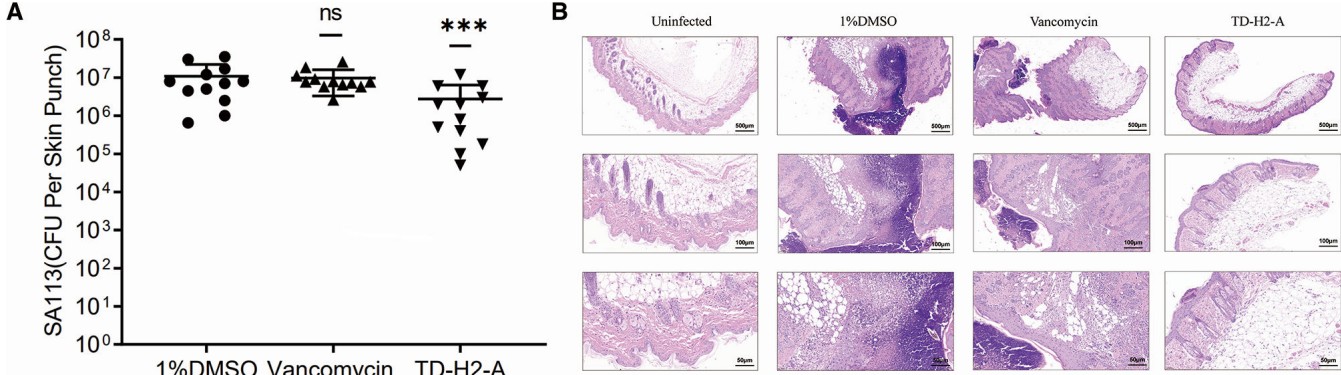

**FIG 5**  (A) SA113 viable bacteria results are shown as mean ± SD; ***$P < 0.001$, and "ns" indicates $P > 0.05$ (unpaired Student's *t*-test and *F* test). (B) Histological examination of skin lesions (The bar represented 50, 100, or 500 µm was indicated.). The arrows in the figure point to where the tissue changed.

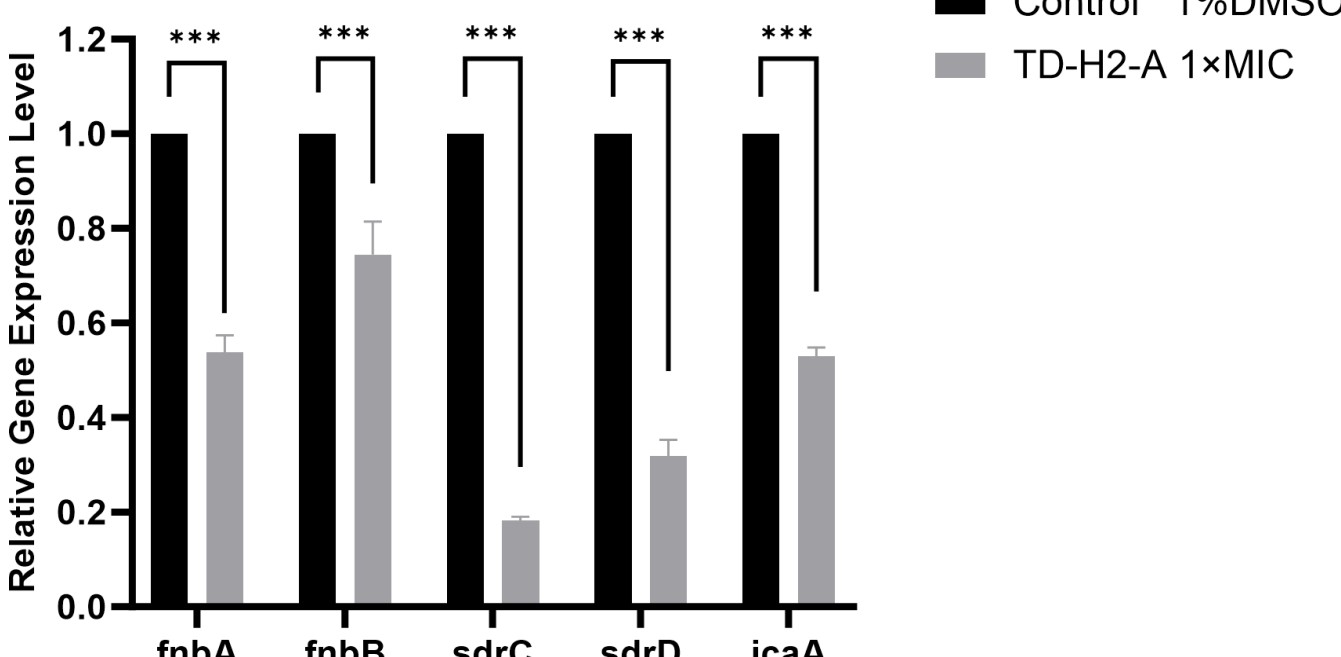

**FIG 6** qRT-PCR results of TD-H2-A's effect on the transcription of biofilm-related genes. All assays were performed with three biologically independent experiments, and the mean ± SD is shown. Statistical differences between control and antibiotic treatment groups were analyzed by one-way ANOVA with Tukey's multiple comparisons test (***$P < 0.001$).

## Effect of TD-H2-A on the expression of Biofilm-related genes

To verify whether TD-H2-A affects the expression of biofilm-related genes, we measured the transcription levels of several biofilm formation-related genes of *S. aureus* SA113 via qRT-PCR. As shown in Fig. 6, when treated with TD-H2-A at 1× MIC, the expressions of *icaA*, *sdrC*, *sdrD*, *fnbA*, and *fnbB*, were significantly downregulated when compared with the control.

## DISCUSSION

*S. aureus* is typically found in the human nasal cavity and skin but it is also able to colonize tissue and artificial surfaces (heart valves, prosthetic orthopedic implants, pacemakers, and vascular catheters), causing many acute and chronic persistent infections (19). And *S. aureus* has become notorious owing to its ability to form a stubborn biofilm and to develop drug resistance (20, 21). *S. aureus* secretes an extracellular polymeric substance that comprises the biofilm matrix (22, 23), which contains one or more surface- and cell-wall-associated proteins, capsular polysaccharides, and extracellular DNA (24). Biofilm-associated infections bring a challenging problem in the clinical field since conventional antibiotic therapies are largely ineffectual (25). It is difficult to eradicate viable bacterial cells embedded in biofilm in the clinical setting (26). The eradication of mature biofilms remains an urgent clinical concern.

Vancomycin and related glycopeptides are antibiotics of last resort for the treatment of severe infections caused by Gram-positive bacteria such as *Enterococcus* species, *S. aureus*, and *Clostridioides difficile* (27, 28), methicillin-resistant *S. epidermidis*, and other multiple antibiotic-resistant infections caused by Gram-positive bacteria but it has little effect on bacteria in biofilms (16, 29). Furthermore, *S. aureus* clinical isolates with reduced susceptibility to vancomycin and, less frequently, complete resistance to vancomycin have emerged in the last 20 years (30–32). Although linezolid and high-dose daptomycin are available for staphylococcal biofilm-associated infections, neither was found to be bacteriocidal against biofilm-embedded bacteria (33). Because *S. aureus* can successfully

colonize and form biofilms in various host environments, such as implant surfaces, the lung, skin, and bone, it is important to consider this when developing anti-infective therapies for these infections. There is currently no single effective treatment for *S. aureus* biofilm infections (19).

In this study, we found that TD-H2-A had potent bacteriocidal activities toward cells in mature biofilms by assessing inhibitory and bacteriocidal activities against *S. aureus* embedded in mature biofilms. It has a bacteriocidal effect not only on *S. aureus* but also on *Enterococcus* (VRE) and *Listeria monocytogenes*. In addition, under the treatment of 10× MIC TD-H2-A, more than 99.4% of the *S. aureus* (SA113) strains in the biofilm were killed. And with 1× MIC TD-H2-A-treated *S. aureus* (SA113), the expression levels of *icaA*, *sdrC*, *sdrD*, *fnbA*, and *fnbB* genes related to biofilm formation have been decreased significantly.

The ideal antimicrobial agent for infection treatment should have low toxicity *in vivo*. Cytotoxicity is a critical factor in assessing any adverse effects of molecules before using them commercially for the prevention and removal of *S. aureus* (34). To advance translational research on potential therapeutic small molecules against *S. aureus*, the compounds must be relatively non-toxic to mammalian cells (35). In this research, MTT results revealed that TD-H2-A had no obvious inhibitory effect on cell proliferation in Vero cell lines. Moreover, it had minimal hemolytic activity on human erythrocytes.

Bacterial cells must keep their membrane intact to survive. When the cell membrane permeability is reduced, the bacterial energy metabolism is affected, and thus drug absorption is reduced, leading to drug resistance (36, 37). *S. aureus* resistance to aminoglycosides, for example, is caused by a decrease in membrane permeability, which results in a decrease in drug intake. Some substances can enhance antibiotic activity by permeating cell membranes (38, 39). In our study, the derivative increased cell membrane permeability at 4× and 8× MICs.

Most notably, we found that the *S. aureus* strain passaged in TD-H2-A showed reduced susceptibility in the resistance experiment, then we performed whole-genome sequencing. According to the sequencing results, six genes were mutated and they identified had no previously known correlation to drug resistance. TD-H2-A has potent activity against multiple resistant Gram-positive pathogens and a low probability of producing detectable resistance, which makes it a promising antibiotic. In addition, using a mouse skin infection model and H&E staining, TD-H2-A showed a better effect against *S. aureus* SA113.

However, the precise antibacterial mechanism of TD-H2-A to *S. aureus* is unclear. TD-H2-A may effectively inhibit *S. aureus* planktonic growth and biofilm formation by targeting the HK-WalK. We hypothesize that it interacts with *S. aureus* WalK-HK to exert its bactericidal effect. We constructed the WalK protein dimer model and the WalK protein-drug small molecule complex molecular docking model. The WalK protein dimer structure model was generated using the protein complex structure prediction tool AlphaFold-multimer (40). To screen for drug-binding sites, possible pockets of the dimer model were calculated by mDPA (41). To further investigate the role of small drug molecules in the ATP binding pocket of the CA domain, Modeler was used to model the monomeric CA domain (42), and AutoDock Vina was used for docking (43). TD-H2-A binds to the CA domain, as well as the HAMP domain. However, the exact mechanism is still unknown. Our research group is currently investigating the underlying molecular mechanism.

## MATERIALS AND METHODS

### Bacterial strains, cells, mice, and compounds

A total of 40 clinical and laboratory strains were obtained from the Second Affiliated Hospital of Nanjing Medical University and the Department of Infection Biology of Eberhard Karls Universität Tübingen. *S. aureus* ATCC25923 was used as the quality control

strain. In addition, human erythrocytes were provided by the authors, and C57BL/6 female mice were purchased from Nanjing Medical University's Animal Core Facility. Vero cells were donated by the research group of the Nephrology Department of Jiangsu Province Hospital. The compound used in this study (TD-H2-A; the purity was 95%) was dissolved in dimethyl sulfoxide (DMSO; Amresco, USA) to 5 mM for use as a stock solution, and TD-H2-A was designed and produced in cooperation with Nanjing Tech University. The source and purity of the TD-H2-A (see supplementary materials). The derivative TD-H2-A structure and systematic name are depicted in Fig. 1.

## MIC determination

*S. aureus*, *Staphylococcus warneri*, *Staphylococcus lugdunensis*, *S. epidermidis*, *Streptococcus pneumoniae*, *Streptococcus constellatus*, *Enterococcus faecalis* VRE366, *Bacillus subtilis*, *Corynebacterium striatum*, *Listeria monocytogenes*, *Klebsiella pneumonia*, *Pseudomonas aeruginosa*, and *Escherichia coli* were used as target strains. Strains were cultured overnight on blood agar plates, and these isolates were identified using the VITEK 2 compact automated microbiology system (bioMérieux, France). The MIC values of TD-H2-A against these strains were determined using the broth microdilution method according to the Clinical and Laboratory Standards Institute guidelines (44).

To determine the MICs of TD-H2-A, a single bacterial colony was incubated at 37 °C under 220 rpm continuous shaking in tubes containing 3 mL Mueller-Hinton broth (MHB, Sigma, Germany) for the middle stage of logarithmic growth (approximately 4 hours). The medium was adjusted with sterile saline until the bacterial suspension density was equal to the 0.5 McFarland standard (~$0.5 \times 10^8$ colony forming unit (CFU)/mL), then diluted to 1:100 into MHB medium. The final concentration of the bacterial solution was about $10^5$ CFU/mL. The compound was serially diluted from 100 µg/mL to 0.06 µg/mL in the 96-well plates. Equal volumes of the bacterial inoculum ($1 \times 10^5$ CFU/mL) were added. The cultures were mixed and placed in a 37°C incubator at 220 rpm for 24 hours. The MIC was defined as the lowest concentration at which no visible microbial growth occurred in the test 96-well plates (44). In addition, a positive control (without compound) and a negative control (without bacterial solution) were set, and quality control for CLSI (Clinical and Laboratory Standards Institute) was performed using *S. aureus* ATCC25923. All experiments were performed in triplicate.

## Cytotoxicity and hemolytic activity assay

The Methyl Thiazolyl Tetrazolium (MTT) assay was used to detect the inhibitory effect of derivatives on cell proliferation (45). To assess the cytotoxicity of TD-H2-A on Vero cells (African green monkey cells), the MTT Cell Proliferation and Cytotoxicity Assay Kit (Protein Bio, Nanjing, China) were used according to the manufacturer's instructions. After culturing in 5% $CO_2$ at 37°C for 24 hours, the cells were harvested and dispensed into 96-well cell culture plates containing $1 \times 10^4$ cells per well in 100 µL and cultured for another 24 hours. TD-H2-A was serially diluted to a series of concentration gradients using multiple proportional dilutions. Next, serial dilutions of six different concentrations (100–3.125 µg/mL) were dispensed into a 96-well plate containing Vero cells. After a 24-hour incubation at 37°C and 5% $CO_2$, the absorbance of the produced formazan dye from Vero cells was measured at 490 nm to determine the mitochondrial activity of viable cells using the MTT method. Cells treated with the solvent (1% DMSO) served as a negative control, cells treated with the nisin served as a positive control, and cells cultured solely in Dulbecco's Modified Eagle Medium served as a blank control. The 1% DMSO showed negligible cytotoxicity to Vero cells. Finally, the results were converted into percentages of cytotoxicity based on the results of the blank control group and treatment groups. A dose-response bar chart of cell viability (%) versus derivative concentration (µg/mL) was plotted. The relative survival rate of cells was T/C%. T was the OD value of the treated cells and C was the OD value of the control cells. Each sample was placed in six wells, and all experiments were performed in triplicate.

The hemolysis of small molecule compounds on healthy human erythrocytes was tested as previously described (46, 47). Red blood cells were washed three times with sterile saline and then diluted to a 5% concentration suspension. The diluted red cell suspensions containing the 1× MIC (6.25 µg/mL) or 4 × MIC (25 µg/mL) of the small molecule compounds were then added to a 96-well plate (200 µL/well) and placed in a 37°C incubator for 1 hour. At the same time, red cell suspensions lacking the small molecule compound TD-H2-A and red cell suspensions containing 1% Triton-100 were used as negative (0% hemolysis) and positive (100% hemolysis) controls, respectively. Thereafter, the treated red cell suspension mixtures were centrifuged at 1,000× *g* for 10 minutes, and 100 µL of the supernatants were transferred to another 96-well plate and measured on a microplate reader at 570 nm. The amount of hemoglobin released from disrupted erythrocytes was determined by measuring the absorbance of the superna-tants at 570 nm. There was no effect of 1% DMSO on erythrocyte morphology, and all experiments were performed in triplicate.

## Resistance development

For the sequential passaging assay, overnight cultures of *S. aureus* (HG001) were diluted (optical density (OD) 600 = 0.02) in 3 mL of TSB medium containing 0.5 MIC TD-H2-A and cultured at 37°C for 24 hours. If bacterial growth was observed after 24 hours, the cultured bacterial solutions were diluted 1:500 into fresh TSB medium supplemented with compound concentration doubling and incubated at 37°C with shaking for another 24 hours. After 25 days of continuous culture with increasing concentrations of TD-H2-A, the final cultures were micro-diluted into MHB broth without antibiotics to determine the MIC. Ofloxacin was used for the control group. We finished the whole-genome sequencing of HG001 treated with 4× MIC TD-H2-A on the 25th day and without treated TD-H2-A.

## Membrane permeability

Cell membrane permeability was assessed by measuring green and red fluorescence using the LIVE/DEAD BacLight Bacterial Viability Kit for microscopy and quantitative assays, following the manufacturer's instructions. This method used two fluorescent nucleic acid stains, SYTO9 and PI, for the drug-treated *S. aureus* strain HG001 cells. In brief, 100 µL of *S. aureus* were incubated with an equal amount of the staining solution containing SYTO 9 dye (10 µM in DMSO solution) and PI (60 µM in DMSO solution) for 15 minutes in the dark. Fluorescence was measured in a microplate reader (Tecan, Infinite M200 PRO, Männedorf, Switzerland) for 5 minutes. TD-H2-A was then added at concentrations of 1×, 2×, 4×, 8 × MIC and measured at different time points (30, 60, 120, and 180 minutes). 1% DMSO was used as a negative control, while nisin was used as a positive control. Fluorescence emissions from these cells were then measured using a multi-label plate reader set to measure emissions at two different wavelengths. The percentage of live versus dead bacteria was indicated by the green versus red ratio.

## Time-killing assay

The time-killing curves of TD-H2-A and vancomycin at concentrations of 2× MIC, 4× MIC, 5× MIC, 6× MIC, 8× MIC, 10× MIC were assayed with *S. aureus* USA300 (~3 × 10$^6$ CFU/mL) suspensions. Time-killing experiments were performed under aerobic conditions at 37°C with shaking at 160 rpm, as previously reported, with some modifications (44, 48). Overnight culture strains were diluted 1:10,000 into fresh tryptic soy broth (TSB) medium and cultured at 37°C with aeration at 160 rpm for 3 hours until mid-exponential phase, then 2× MIC, 4× MIC, 5× MIC, 6× MIC, 8× MIC, 10× MIC corresponding TD-H2-A and vancomycin were added to the treatment group, respectively. Bacteria not treated with antibiotics served as a control. At different time points (2, 4, 6, 8, and 24 hours), aliquots (0.1 mL) were removed and serially diluted with phosphate-buffered saline (PBS). Furthermore, 100 µL of bacteria was evenly coated on tryptic soy agar (TSA) plates

and cultured at 37°C for 24 hours. Bacterial colonies were counted and determined by plotting colony counts (CFU/mL) against time. The results were independently presented in triplicate as the mean and standard deviations (SD).

## Activity of killing bacteria embedded in mature biofilms of thiazolidione derivative

An overnight *S. aureus* SA113 culture was diluted 1:200 into a 0.25% glucose-containing TSB medium. Then the biofilm was formed for 24 hours (18). Aliquots of the inhibitors at 1×, 5×, and 10× MIC concentrations were mixed with the same volume of the bacterial cultures in TSB medium, added in triplicate to a 12-well polystyrene plate (1 mL/well), and statically incubated for 16 hours. As a positive control, wells with no derivatives were used, and 10× MIC vancomycin was added to the treatment group. The wells were gently washed three times with PBS and sonicated for 20 seconds in an ultra-sonication bath, then the diluted bacteria were inoculated onto TSA plates and incubated at 37°C for 24 hours to count bacterial CFU. All experiments were performed in triplicate.

## Determination of viability of mature biofilm-embedded cells by confocal laser scanning microscopy

The effect of thiazolidione derivatives on cell viability in mature biofilms (24 hours) was determined using the Live/Dead Bacterial Viability method (Live/Dead BacLight, Molecular Probes, Eugene, Oregon, USA) with SYTO 9 and propidium iodide (PI) dyes (1 mL/well) to stain live and dead cells within mature biofilms (49). The *S. aureus* (SA113) bacterial liquid in the growth mid-log phase was adjusted with sterile saline until the bacterial suspension density was equal to a 0.5 McFarland standard (~$0.5 \times 10^8$ CFU/mL), then diluted at a ratio of 1:100 and inoculated in TSB medium. The final concentration of the bacterial solution was about $10^5$ CFU/mL. The diluted bacteria were added to a 6-well plate (2 mL/well), mixed, and incubated at 37°C for 24 hours; planktonic bacteria were then removed and discarded. The wells were gently washed three times with PBS, and fresh TSB containing TD-H2-A monotherapy (at 1×, 5×, and 10 × MICs) was then added and incubated at 37°C for another 12–16 hours. After staining for 20 minutes, mature biofilms were examined using a Leica confocal laser scanning microscope (CLSM, Leica) with an oil immersion objective. Viable cells in the biofilms exhibited green fluorescence, while dead cells exhibited red fluorescence. The percentages of live bacteria in the total bacterial counts were calculated by ImageJ software (Rawak Software Inc., Stuttgart, Germany). This assay was performed in triplicate and yielded comparable results (44).

## Animal experiments

A mouse model of skin and soft tissue infection induced by S. aureus SA113 subcutaneous inoculation was used (50–52). All mice (*n* = 42, 14 for each group, 8-week old) were placed in a room with strictly controlled conditions (a temperature of 23 ± 3°C, a relative humidity of 60% ± 5%, and a 12/12 hour light-dark cycle) and fed commercially available standard chow diets and neutral water *ad libitum* at all times. There was one control group (1% DMSO control group) and two treatment groups (TD-H2-A and vancomycin). Each mouse was weighed prior to inoculation (all mice weighed about 17–21 g). The mice were anesthetized with an intraperitoneal injection of 1% sodium pentobarbital (50 mg/kg). At this point, mice were identified with ear punches to allow tracking and identification of individual animals. Hair was removed from the inoculation site to allow for better visualization and more accurate measurements of the abscess. The inoculation site, typically in the central dorsal region, was carefully shaved with an electric clipper. The skin over the intended inoculation site was disinfected with 70% ethanol. After that, a 1 cm wound was cut with sterile scissors, and the back of each mouse was covered with a 7-mm filter paper containing 20 μL of $1 \times 10^8$ CFU/mL *S. aureus*, which was then tightly applied with medical tape. The anesthetized mice recovered in a warm environment and then returned to their home cage. The mice were then injected subcutaneously in the

dorsal region with 20 µL of 1% DMSO, the inhibitor TD-H2-A (at a $5 \times$ MIC concentration), and vancomycin (10 µg/mL) at 24-hour intervals for three consecutive days. After 6 hours of the third injection, the mice were sacrificed, and their infected skin tissues were removed. After 30 seconds of washing with sterile saline, two infected skin tissues from each group were fixed with formaldehyde and sent for pathological sections. The remaining 12 skin tissues in each group were ground with grinding rods, diluted in multiple ratios, and inoculated onto blood agar plates. Thereafter, the tissues were incubated at 37°C for 24 hours, and bacterial CFU counts were calculated using GraphPad Prism 9.0 (GraphPad Software Inc., San Diego, CA, United States). All experiments were performed in triplicate.

## Histological analysis

The mice were sacrificed by administering three times the anesthetic dose of sodium pentobarbital intraperitoneally. The wound samples, including wound beds and healthy skin 2 mm from the wounds' peripheral edges, were collected using a dissecting scissor. The skin specimens were fixed overnight in a 10% formaldehyde solution, embedded in paraffin, and cut into 5-µm-thick sections. The sections were stained with hematoxylin and eosin (H&E) for histological examination. Slides were deparaffinized in a xylene bath, dehydrated with graded ethanol (100%, 95%, 80%, and 70%), washed in distilled water, and stained with H&E. Finally, the slices were sealed with neutral gum. Histological images were captured at a magnification of 0–400× using a scanner (Pannoramic MIDI, 3DHIS TECH, Hungary) and analyzed using the Case Viewer software (Version 2.4.0.53492, 3DHIS TECH Ltd., Budapest, Hungary) (53). For each experiment, two samples were collected from each group, and the experiment was repeated three times.

## RNA extraction and reverse transcription-quantitative PCR

*S. aureus* SA113 strains were cultured in TSB at 37°C for 12 h. The above overnight culture of *S. aureus* SA113 strain was inoculated at 1:100 in 3 mL of TSB medium. After approximately 4 h of 37°C 200 rpm shaking incubation, TD-H2-A of $1 \times$ MIC were added for 15 min. Positive control was the tubes where bacteria were inoculated with 1% DMSO. RNA extraction was performed according to the manufacturer's instructions (EASYspin Plus Bacterial RNA Rapid Extraction Kit, Proteinbio, Nanjing). Next, extracted RNA (1 µg of each) was used as the template for cDNA synthesis using a TRUEscript RT MasterMix (OneStep gDNA Removal) (Proteinbio, Nanjing). Quantitative real-time PCR (qRT-PCR) was performed using the $2 \times$ Sybr Green (Proteinbio, Nanjing) and Roche LightCycler 480 Fluorescence quantitative PCR instrument. The primer pairs used for qRT-PCR are shown in Table 2, with *gyrB* as the constitutively expressed gene. The data were analyzed using a previously described relative quantitative ($2^{-\Delta\Delta Ct}$), fold changes in gene expression were

**TABLE 2** Primer sequences for quantitative RT-PCR

| Gene | Primer |
|------|--------|
| *icaA* | Forward 5′-CTGGCGCAGTCAATACTATTTCGGGTGTCT-3′ |
| | Reverse 5′-GACCTCCCAATGTTTCTGGAACCAACATCC-3′ |
| *sdrC* | Forward 5′-AAGTGGTCATGAAGCTAAAGCGGC-3′ |
| | Reverse 5′-CTGATCTGCAGTTGCAGTTTGCGT-3′ |
| *sdrD* | Forward 5′-GCAGATGGTGGCGAAGTTGACG-3′ |
| | Reverse 5″-CACTGTCTGAGTCTGAGTCGCTGT-3′ |
| *fnbA* | Forward 5′-GACCCGCTTCACTAT-3′ |
| | Reverse 5′-ACACCGCTTGACATT-3′ |
| *fnbB* | Forward 5′-AATAAGGATAGTATGGGTAG-3′ |
| | Reverse 5′-CACAAGTAATGGTCGGT-3′ |
| *gyrB* | Forward 5′-CCAGGTAAATTAGCCGATTGC-3′ |
| | Reverse 5′-ATCGCCTGCGTTCTAGAGTC-3′ |

calculated and RNA transcription levels of biofilm-related genes were obtained. Three biological replicates and three technical replicates were performed for each gene tested.

## Whole-genome sequencing

We analyzed the whole-genome sequencing of HG001 treated with 4× MIC TD-H2-A on the 25th day and not treated with TD-H2-A. Sequencing was performed by Shanghai Biozeron Biotechnology Co. Ltd (Shanghai, China). For Illumina pair-end sequencing of each strain, at least 1 µg genomic DNA was used for sequencing library construction. Paired-end libraries with insert sizes of ~400 bp were prepared following Illumina's standard genomic DNA library preparation procedure. Purified genomic DNA is sheared into smaller fragments with a desired size by Covaris, and blunt ends are generated by using T4 DNA polymerase. After adding an "A" base to the 3′ end of the blunt phosphorylated DNA fragments, adapters are ligated to the ends of the DNA fragments. The desired fragments can be purified through gel-electrophoresis, then selectively enriched and amplified by PCR. The index tag could be introduced into the adapter at the PCR stage as appropriate and we did a library quality test. The sample was sequenced by the Illumina NovaSeq 6000 platform (150 bp*2, Shanghai Biozeron Biotechnology Co., Ltd, Shanghai, China). There were three replicates sequenced in this experiment.

## Statistical analysis

GraphPad Prism 9.0 (GraphPad Software Inc., San Diego, CA, United States) and Statistical Package for the Social Sciences software (SPSS Inc., Chicago, IL, United States) were used to statistically analyze the data and create graphs. The unpaired Student's $t$-test was used to compare data, and the $F$ test and ANOVA test were used to compare variances. Furthermore, a $P$-value of less than 0.05 was considered statistically significant for all analyses. All experiments were performed in triplicate.

## ACKNOWLEDGMENTS

We would like to express our sincere gratitude to the Wutai Lab Center of the Second Affiliated Hospital of Nanjing Medical University and thank Fengxia He for her assistance with the pathological analysis. We also thank Prof. Xiang Yu of Shanghai Jiao Tong University and Pei Li of Nanjing Agricultural University for their help in the genome sequence analysis.

This work was supported by the National Natural Science Foundation of China (grant no. 81802071 and 81871622) and Open Subject youth program of Key Laboratory of Medical Molecular Virology (MOE/NHC/CAMS) of Fudan University (grant no. FDMV-2023004).

Y.Z. and S.H. designed the studies; Y.Z. and D.Q. obtained funding; R.Z., B.D., Y.L., and F.X. performed the experiments; H.W. contributed to the concept of the study and critical revision; R.Z. and B.D. wrote the manuscript; S.H. contributed the strains and CLSM. All authors read and approved the submitted version.

## AUTHOR AFFILIATIONS

[1]Laboratory Medicine Center, The Second Affiliated Hospital, Nanjing Medical University, Nanjing, China

[2]College of Biotechnology and Pharmaceutical Engineering, Nanjing Tech University, Nanjing, China

[3]Department of Microbial Genetics, Interfaculty Institute of Microbiology and Infection Medicine, University of Tübingen, Tübingen, Germany

[4]Key Laboratory of Medical Molecular Virology (MOE/NHC/CAMS) School of Basic Medical Sciences, Shanghai Medical College, Fudan University, Shanghai, China

[5]Department of Infection Biology, Interfaculty Institute of Microbiology and Infection Medicine, University of Tübingen, Tübingen, Germany

## AUTHOR ORCIDs

Bingyu Du ⓘ http://orcid.org/0009-0009-8703-5591
Simon Heilbronner ⓘ http://orcid.org/0000-0002-6774-2311
Yanfeng Zhao ⓘ http://orcid.org/0000-0001-6475-6171

## ETHICS APPROVAL

All procedures on mice were conducted in accordance with relevant national and international guidelines (the Regulations for the Administration of Affairs Concerning Experimental Animals, China) and were approved by the Institutional Animal Care and Use Committee (IACUC) of Nanjing Medical University (IACUC animal project no. 2110021).

## ADDITIONAL FILES

The following material is available online.

### Supplemental Material

**Supplemental material (Spectrum02327-23-s0001.pdf).** Synthetic method for TD-H2-A, Table S1, and Figure S1.

### Open Peer Review

**PEER REVIEW HISTORY (review-history.pdf).** An accounting of the reviewer comments and feedback.

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
