## [Reviewer comments · Microbiology Spectrum]

Microbiology Spectrum

Antimicrobial and Anti-biofilm Activity of A Thiazolidinone Derivative against *Staphylococcus aureus* *in vitro* and *in vivo*

Rui Zhao, Bingyu Du, Yue Luo, Fen Xue, Huanhuan Wang, Di Qu, Shiqing Han, Simon Heilbronner, and Yanfeng Zhao

Corresponding Author(s): Yanfeng Zhao, 南京医科大学附属第二医院

Review Timeline:

Submission Date:	June 4, 2023
Editorial Decision:	August 17, 2023
Revision Received:	November 8, 2023
Editorial Decision:	December 18, 2023
Revision Received:	January 8, 2024
Accepted:	January 11, 2024

Editor: Tomefa Asempa

Reviewer(s): Disclosure of reviewer identity is with reference to reviewer comments included in decision letter(s). The following individuals involved in review of your submission have agreed to reveal their identity: Hassan M Al-Tameemi (Reviewer #1); Salwa E. Gomaa (Reviewer #2); sreekanth reddy basireddy (Reviewer #3)

Transaction Report:

DOI: <https://doi.org/10.1128/spectrum.02327-23>

August 17, 2023

Dr. Yanfeng Zhao
南京医科大学附属第二医院
Nanjing
China

Re: Spectrum02327-23 (Antimicrobial and Anti-biofilm Activity of A Thiazolidinone Derivative against *Staphylococcus aureus*)

Dear Dr. Yanfeng Zhao:

Link Not Available

Sincerely,

Tomefa Asempa

Journals Department
Reviewer comments:

Reviewer #1 (Comments for the Author):

Dear Editor ,

Thank you for including me as a reviewer for the manuscript by Zhao et al., (Antimicrobial and Anti-biofilm Activity of A Thiazolidinone Derivative against *Staphylococcus aureus*). The manuscript discusses the development of the TD-H2-A derivative as a potential antimicrobial. The authors investigated the bactericidal and anti-biofilm activities of the thiazolidinone derivative (TD-H2-A) against *S. aureus*. 40 non-duplicate strains were collected, and the minimum inhibitory concentrations (MICs) of TD-H2-A were determined. The compound was found to have strong bactericidal activity against mature biofilms and low cytotoxicity and hemolytic activity against Vero cells and human erythrocytes. TD-H2-A also showed a good bactericidal effect on *S. aureus* SA113-infected mice. The authors concluded that TD-H2-A could be a potential agent to combat biofilm infections and multidrug-resistant staphylococcal infections.

Major comments:

1. Regarding the synthesized TD-H2-A, what was the purity of the compound obtained?
2. Figure 2 A: Line 124: The survival rates of Vero cells after treatment with different concentrations ... Why the 12.5-25 Concentrations ($\mu\text{g}/\text{mL}$) had higher survival rates compared to 3.125-100 ($\mu\text{g}/\text{mL}$)? Are these differences statistically significant?
3. Line 129: The hemolytic effect of the new synthetic derivatives was assessed on healthy human erythrocytes. "Concentration below the MIC did not have an obvious hemolytic effect on the hemolysis ratio of human red blood cells, which was less than 0.67%. At a higher concentration of 25 $\mu\text{g}/\text{mL}$, the hemolysis ratio of the TD-H2-A derivative was less than 1.42%(data not shown)". Is there any reason to use up to 25 $\mu\text{g}/\text{mL}$ concentration for testing hemolysis effect and not up to 100 $\mu\text{g}/\text{mL}$ like in the survival rate experiment of Vero cells in Figure 2-A? I feel the hemolysis data are important enough to be presented in a figure.
4. Line 145: "By comparing the whole genome sequencing of HG001 treated with 4mic TD-H2-A on the 25th day and without treated TD-H2-A(Table S1), we found that two genes had a single-base mutation in *mnhD1*(Na^+/H^+ antiporter *Mnh1* subunit D) and a multi-base insertion mutation in *TrkA* family potassium uptake protein, while it was not reported that the correlation between the two genes and drug resistance". There is a lack of information about the genome sequencing method and the number of cultures sequenced. How many replicates were sequenced? There is a possibility of discovering more or other mutations if all the treatment cultures (replicates) were sequenced.
5. Figure 3 A-F. The period of time points 8-24 hours is too long to be left without measuring OD.... What if there is growth improvement or decline during this unmeasured period of time? Because of that, the graph is squeezed and does not show the usual growth pattern of *S. aureus* in the control group. I think removing the points 8-24 or adding measurement points in between 8-24 hrs is helpful.
6. Figure 3 F: Growth at 24 hours seems higher than at the 8 hrs point. Does it mean suppressors or mutations are starting to appear at this point?
7. Line 183 : "The effects of the derivative on cell viability in mature *S. aureus* biofilms were detected using a CLSM with Live/Dead staining. In the 10 \times TD-H2-A treatment group, more than 99.4% of the bacteria in the biofilm were stained red by PI, indicating that the majority of the bacteria were dead (Fig. 3C)". How come the number of total cells ((Dead (red cells) and green (live cells)) in the TD-H2-A 1XMIC treatment is less than the total green cells in the control 1%DMSO or total red cells in the TD-H2-A 10 XMIC treatment groups?!
8. Line 199: H&E staining revealed that *S. aureus* SA113 induced infection in mouse skin tissues, whereas control mice receiving 1% DMSO had obviously positive staining. Needs further elaboration/Explanation.

Minor comments:

- Language and typos: The English should be revised all throughout the manuscript for clarity and correct usage.
- Line 262. Because TD-H2-A had little effect on gram-negative bacteria.. The statement is confusing .
- Line 171: The strains in the control group had been steadily growing (Fig.3A). I think this should be "The strains in the control group had been steadily growing (Fig.3A-F)"
- Figure 4-B Line 519 : Photos need numbering and referencing in the text.

Reviewer #2 (Comments for the Author):

The authors investigated the bactericidal and anti-biofilm activities of thiazolidinone derivative (TD-H2-A) against *S. aureus*. Overall, this is a clear, concise, manuscript. The introduction has sufficient information. The methods are generally appropriate, the results and discussion are clear and strongly supported by the data, here are my concerns:

- 1- I strongly recommend adding a graphical abstract to the manuscript.
- 2- For the title it would be better to add (in vitro and in vivo approach) to the end of it.
- 3- Thiazolidinone-based molecules are attractive targets in the rational design of drug-like compounds, which possess anti-inflammatory, antioxidant, antitumor, choleric, diuretic, and other activities. In the current study, the authors repurposed them as antibiofilm and antibacterial agent, there is a lack of information about drug repurposing, the authors need to add short sentences to the introduction section about this approach as a new alternative approach to fight against antimicrobial resistance and the advantages of this approach over synthesis of new antibiotics. You can also support your text with these papers that use this approach;

*Kocking down *Pseudomonas aeruginosa* virulence by oral hypoglycemic metformin nano emulsion

*Alleviating the virulence of *Pseudomonas aeruginosa* and *Staphylococcus aureus* by ascorbic acid nanoemulsion

4- It is better to mention from which patients the clinical isolates used in this study were obtained (with infection of the urogenital tract, diabetic foot, etc.).

5- In mice model, 14 mice/group is really a huge number and unethically and 6-8 mice/group should be sufficient for statistics. In addition, please clarify in the methodology section why the treatment lasts only for 3 days. Was this being enough for the healing

process?

6- It would be really great if the authors study the effect of TD-H2-A on the expression level of genes controlling biofilm in *S. aureus* using qRT-PCR experiment. This will support your findings.

Reviewer #3 (Comments for the Author):

In this study, the authors investigated the bactericidal and anti-biofilm activity of compound TD-H2-A, a derivative of thiazolidine, in various staphylococcus aureus strains. Repurposing of non-antibiotic group of drugs and their derivatives for the treatment of resistant bacterial infections has been a promising approach especially in the era of emergence of Multidrug resistant organisms like MRSA, VRE, and carbapenem resistant GNB. *S. aureus* is one of the most commonly isolated bacteria from clinical infections which is associated with both acute and chronic infections and has the ability to form biofilms and develop drug resistance. MRSA isolates associated with biofilm formation are hard to eradicate, resulting in high morbidity and mortality. Compounds with both antibacterial and antibiofilm activity, which are less prone to develop resistance are the much-needed drugs for these infections.

In the present study, the new thiazolidine derivative TD-H2A has shown good bactericidal and anti-biofilm activity against various clinical and reference strains of *S. aureus* and other bacteria. The authors performed a series of experiments including MIC detection, time kill assays, and confocal laser scanning microscopy with live/dead cell staining for demonstrating the drug activity. Serial passaging experiments were conducted with increasing concentrations of the drug along with WGS sequencing to understand the kinetics of resistance development in *S. aureus*. The authors also evaluated the safety profile of the drug by investigating the cytotoxicity on Vero cells by using MTT assay and hemolytic properties on human RBC. Mouse skin infection model was used to evaluate the pathology by using H&E stain as well as detecting the viable bacterial counts. All these experiments being the strengths of the study, not knowing the precise mechanism of action of the compound is its major drawback.

Though It is a well conducted research study, the author might need to address the following minor issues

1. In the abstract (line 31) it is mentioned that a total of 40 non duplicate *S. aureus* were included in the study.. but the authors included other staphylococcal species as well as other bacteria from different genera for evaluation (Table1). It should be corrected.
2. Only Fig 1 legend is shown below the figure. Legends are missing for all the other figures.
3. Line 146 - mic should be capitalized
4. Line 151- figure 3c should be replaced by figure 2c as the membrane permeability at different time points is shown in fig 2c.
5. Line 169 - you find living CFU...should be replaced by appropriate words
6. Line 212- nasal cavity should be replaced by nasalcavity
7. Line 227- no little effect - should be either no effect or little effect
8. Line 241- mic90 should be replaced by MIC90 and valve should be replaced by value
9. Line 314- it is mentioned that no visible growth observed in the test tubes.. As per the methodology, MIC testing was done by broth microdilution method in 96 well plates... discrepancies should be corrected.
10. Line 396 &397 - It is preferable to keep the heading in bold letters.

Reviewer #4 (Comments for the Author):

The manuscript by Zhao et al. continues from previous work in the development of an antimicrobial compound that can be effective against *Staphylococcus aureus* within a biofilm. The authors demonstrate that their new derivative, TD-H2-A, has both bactericidal and anti-biofilm activity.

Recommendations:

1. The authors should explain in the manuscript why *S. aureus* HG001 was used in some experiments while *S. aureus* SA113 was used in other experiments.
2. The term MIC90 is misused in the manuscript. MIC90 refers to the concentration of a compound which inhibits 90% of the tested bacterial strains, and it is best practice that data from at least 10 strains of a single species are used to calculate the MIC90 The term MIC should be used instead on lines 39, 112, 241, and in Table 1. As written, the results section does not flow particularly well. The primary reason for this is that the individual sub-sections feel disjointed from one another. This could be remedied with the addition of sentences at the beginning and end of each sub-section, which would serve to provide context and justification for performing the experiments and also serve as bridges between the sub-sections.
3. Line 31: Table 1 indicates that the strain collection consisted of 20 *S. aureus* strains and 20 strains encompassing many other genera and species, not 40 non-duplicate *S. aureus* strains, as stated on line 31.
4. Abstract: The target of TD-H2-A should be stated in the abstract.
5. Lines 74-75: It is not accurate that the biofilm matrix restricts penetration of host defense molecules and antimicrobial agents. Many molecules are able to penetrate the full depth of biofilms but are inactive due to bacterial cell metabolic state or other reasons.
6. Paragraph beginning line 110: The authors should define what they consider to be "potent antimicrobial activity" (line 114-

115) of the TD-H2-A compound.

7. Lines 145-150 are difficult to understand as written and could benefit from revision. In addition, the authors should clarify: (i) What does 4mic mean? Do the authors mean to say 4x MIC?, (ii) What does "without treated TD-H2-A" mean? Was this the HG001 isolate from Day 0?, (iii) Line 147 says two genes had a single base mutation in the *mnhD1* gene, which does not make sense and is not consistent with data in the supplemental table., (iv) The insertion mutation was in a gene, not a protein (line 149).

8. Line 151: Should this be Figure 2C, not Figure 3C?

9. Lines 173-180: The p values on lines 175 and 178 do not make sense with the comparisons as written in this section. Is it possible the p values are misplaced?

10. Line 191: This is a subjective assessment and should be removed or replaced with a quantitative assessment.

11. Line 194: A few sentences explaining the mouse infection model is needed at the start of the "Mouse infection model and pathological sections" sub-section.

12. The sentence on lines 199-200 does not make sense as written. Are the control mice that received 1% DMSO a control for the infection (i.e., received no *S. aureus*) or for the treatment (i.e., were infected with *S. aureus* but received only vehicle)?

13. Sentence on lines 221-222: Reference 24 is not an appropriate citation for this statement.

14. Paragraph on lines 237-244: This paragraph is confusing as written, as sentences about the activity of TD-H2-A against *S. aureus* biofilms are interrupted with statements about the MIC of the compound when tested against other genera. On line 244, please specify which bacterial species it was in which 99.4% of the biofilm bacteria were killed.

15. Lines 261-263: The authors did not show that increased cell permeability increased drug absorption, and it is unclear how the following statement about the TD-H2-A having little effect on Gram-negative bacteria is related to cell permeability.

16. Lines 264-269: The information in this paragraph contradicts lines 138-150, Figure 2B, and the supplemental table, where it is indicated that the *S. aureus* strain passaged in TD-H2-A showed reduced susceptibility and gained two mutations.

17. Line 273: The authors should be more specific than saying "According to the aforementioned methods".

18. Line 292: The authors should provide details on how the TD-H2-A compound was obtained for use in this study.

19. The sentence on L293-294 needs to be revised, "The derivatives' structures and systematic names are depicted in Figure 1." This sentence appears to have been pulled from a previous study where multiple derivatives were made and examined, whereas here only a single compound is studied.

20. Line 316: In what way was *S. aureus* ATCC25923 used for quality control in the MIC assays? Was it tested with a particular antibiotic? Quality control cannot be assured if a known drug/agent is not used.

21. Line 362: Details of whole genome sequencing methods are needed. The genome sequence results need to be deposited in a repository for public access.

22. Lines 396-397: Should these lines be formatted as a section header?

23. In L405 "ultrasonically" should be changed to sonicated. Line 405: What volume and liquid were used for the sonication step? Was sonication performed with a probe or bath sonicator?

24. Line 470, Statistical analysis paragraph: One or more figure legends indicate that an ANOVA test was used, but it is not mentioned in this paragraph.

25. The Figure 2 and Figure 3 legends each contain several lines of text summarizing the results. This text should be moved to the results and deleted from the legends.

26. Figures 2 and 4 legends: Please state the statistical test used in each figure.

27. Line 506: There is no blue line in the graph. The vancomycin treatment data are shown as red or pink.

28. L507 "**** indicates $p < 0.01$, *** $p < 0.001$," ** is not used in the referenced figure (Fig. 3B), and should be removed from the figure description. Additionally, an explanation of what **** indicates should be added here as it is utilized in the figure, but not explained in the figure description.

29. On L518 "**** $p < 0.01$ " should be removed as it is not used in the referenced figure (Fig. 4A). Additionally, an explanation should be added to this figure legend to explain the meaning of "NS".

30. In Fig. 2B the results of only a single experiment are shown. If possible, the results of all three replicates should be combined and shown here.

31. In Figs. 2C and 3A the concentrations are listed as "5*MIC", whereas within the text and Fig. 3C the concentrations are listed as "5x MIC". The labeling in Fig. 3 should be changed to be in line with the rest of the manuscript.

32. There is a discrepancy between the labeling of Figs. 2C, 3B, and 4A. Figs. 2C and 3B use "ns" to indicate a p-value > 0.05 , while Fig. 4A uses "NS". This labeling should be made consistent across figures.

33. Table 1: Why is the MIC listed as 25.0 for some strains but 25.04 for other strains? In addition, why is MBC data given in the table footnote? No methods describing MBC have been included in the manuscript.

34. Supplementary table: Several columns are empty and should be removed. Give the definitions of the abbreviations used in the headers for the columns that do contain data. In the table title, does '4mic' mean '4xMIC'?

35. Figure 2A x-axis should be labeled as "Agent Concentration". Figure 2C y-axis should be labeled as "Green/Red Ratio".

36. The labeling of the six graphs as lowercase a-f in Figure 3A is confusing.

37. The order of strains should be the same in Figures 4A and 4B, i.e., 1% DMSO on the left, vancomycin in the middle, TD-H2-A on the right.

38. Significant attention should be given to the writing of the manuscript, as there are currently multiple typos and grammatical errors throughout. In addition, the following specific revisions are needed for improved clarity or accuracy:

a) Lines 56-60 is a long sentence.

b) Line 72 need to add word to read "*S. aureus* infections."

c) Lines 124-129 contain redundant information.

- d) Line 169 revise remove informal writing "you find living"
- e) Line 174 revise to remove informal writing "can hardly be"
- f) Line 191 revise to remove informal writing "mostly died" and "basically died"
- g) Lines 212-216 should be broken into shorter sentences.
- h) Line 212 nasal, not nosal
- i) Line 217 need to revise to read "that comprises the biofilm matrix"
- j) Line 225: Replace Clostridium with Clostridioides.
- k) The first sentence in the paragraph on line 253 is unnecessary.
- l) Line 276 revise to remove informal writing "But we yet don't know"
- m) Line 297: VRE should be replaced with the bacterial genus and species.
- n) Line 331: Revise to say "cells treated with nisin..."

Staff Comments:

Preparing Revision Guidelines

Please return the manuscript within 60 days; if you cannot complete the modification within this time period, please contact me. If you do not wish to modify the manuscript and prefer to submit it to another journal, please notify me of your decision immediately so that the manuscript may be formally withdrawn from consideration by Microbiology Spectrum.

Dear Editor ,

Thank you for including me as a reviewer for the manuscript by Zhaoa et al., (Antimicrobial and Anti-biofilm Activity of A Thiazolidinone Derivative against *Staphylococcus aureus*). The manuscript discusses the development of the TD-H2-A derivative as a potential antimicrobial. The authors investigated the bactericidal and anti-biofilm activities of the thiazolidinone derivative (TD-H2-A) against *S. aureus*. 40 non-duplicate strains were collected, and the minimum inhibitory concentrations (MICs) of TD-H2-A were determined. The compound was found to have strong bactericidal activity against mature biofilms and low cytotoxicity and hemolytic activity against Vero cells and human erythrocytes. TD-H2-A also showed a good bactericidal effect on *S. aureus* SA113-infected mice. The authors concluded that TD-H2-A could be a potential agent to combat biofilm infections and multidrug-resistant staphylococcal infections.

Major comments:

1. Regarding the synthesized TD-H2-A, what was the purity of the compound obtained?
2. Figure 2 A: Line 124: The survival rates of Vero cells after treatment with different concentrations ... Why the 12.5–25 Concentrations ($\mu\text{g}/\text{mL}$) had higher survival rates compared to 3.125–100 ($\mu\text{g}/\text{mL}$)? Are these differences statistically significant?
3. Line 129: The hemolytic effect of the new synthetic derivatives was assessed on healthy human erythrocytes. "Concentration below the MIC did not have an obvious hemolytic effect on the hemolysis ratio of human red blood cells, which was less than 0.67%. At a higher concentration of 25 $\mu\text{g}/\text{mL}$, the hemolysis ratio of the TD-H2-A derivative was less than 1.42%(data not shown)". Is there any reason to use up to 25 $\mu\text{g}/\text{mL}$ concentration for testing hemolysis effect and not up to 100 $\mu\text{g}/\text{mL}$ like in the survival rate experiment of Vero cells in Figure 2-A? I feel the hemolysis data are important enough to be presented in a figure.
4. Line 145: "By comparing the whole genome sequencing of HG001 treated with 4mic TD-H2-A on the 25th day and without treated TD-H2-A(Table S1), we found that two genes

had a single-base mutation in *mnhD1* (Na^+/H^+ antiporter Mnh1 subunit D) and a multi-base insertion mutation in TrkA family potassium uptake protein, while it was not reported that the correlation between the two genes and drug resistance". There is a lack of information about the genome sequencing method and the number of cultures sequenced. How many replicates were sequenced? There is a possibility of discovering more or other mutations if all the treatment cultures (replicates) were sequenced.

5. Figure 3 A-F. The period of time points 8-24 hours is too long to be left without measuring OD.... What if there is growth improvement or decline during this unmeasured period of time? Because of that, the graph is squeezed and does not show the usual growth pattern of *S. aureus* in the control group. I think removing the points 8-24 or adding measurement points in between 8-24 hrs is helpful.
6. Figure 3 F: Growth at 24 hours seems higher than at the 8 hrs point. Does it mean suppressors or mutations are starting to appear at this point?
7. Line 183 : "The effects of the derivative on cell viability in mature *S. aureus* biofilms were detected using a CLSM with Live/Dead staining. In the 10 \times TD-H2-A treatment group, more than 99.4% of the bacteria in the biofilm were stained red by PI, indicating that the majority of the bacteria were dead (Fig. 3C)". How come the number of total cells ((Dead (red cells) and green (live cells)) in the TD-H2-A 1XMIC treatment is less than the total green cells in the control 1%DMSO or total red cells in the TD-H2-A 10 XMIC treatment groups?!
8. Line 199: H&E staining revealed that *S. aureus* SA113 induced infection in mouse skin tissues, whereas control mice receiving 1% DMSO had obviously positive staining. Needs further elaboration/Explanation.

Minor comments:

- Language and typos: The English should be revised all throughout the manuscript for clarity and correct usage.

- Line 262. Because TD-H2-A had little effect on gram-negative bacteria.. The statement is confusing .
- Line 171: The strains in the control group had been steadily growing (Fig.3A). I think this should be "The strains in the control group had been steadily growing (Fig.3A-F)"
- Figure 4-B Line 519 : Photos need numbering and referencing in the text.

The authors investigated the bactericidal and anti-biofilm activities of thiazolidinone derivative (TD-H2-A) against *S. aureus*. Overall, this is a clear, concise, manuscript. The introduction has sufficient information. The methods are generally appropriate, the results and discussion are clear and strongly supported by the data,

here are my concerns:

- 1- I strongly recommend adding a graphical abstract to the manuscript.
- 2- For the title it would be better to add (*in vitro* and *in vivo* approach) to the end of it.
- 3- Thiazolidinone-based molecules are attractive targets in the rational design of drug-like compounds, which possess anti-inflammatory, antioxidant, antitumor, choleric, diuretic, and other activities. In the current study, the authors repurposed them as antibiofilm and antibacterial agent, there is a lack of information about drug repurposing, the authors need to add short sentences to the introduction section about this approach as a new alternative approach to fight against antimicrobial resistance and the advantages of this approach over synthesis of new antibiotics. You can also support your text with these papers that use this approach;

*Kocking down *Pseudomonas aeruginosa* virulence by oral hypoglycemic metformin nano emulsion

*Alleviating the virulence of *Pseudomonas aeruginosa* and *Staphylococcus aureus* by ascorbic acid nanoemulsion

- 4- It is better to mention from which patients the clinical isolates used in this study were obtained (with infection of the urogenital tract, diabetic foot, etc.).
- 5- In mice model, 14 mice/group is really a huge number and unethically and 6-8 mice/group should be sufficient for statistics. In addition, please clarify in the methodology section why the treatment lasts only for 3 days. Was this being enough for the healing process?

6- It would be really great if the authors study the effect of TD-H2-A on the expression level of genes controlling biofilm in *S. aureus* using qRT-PCR experiment. This will support your findings.

In this study, the authors investigated the bactericidal and anti-biofilm activity of compound TD-H2-A, a derivative of thiazolidine, in various *staphylococcus aureus* strains. Repurposing of non-antibiotic group of drugs and their derivatives for the treatment of resistant bacterial infections has been a promising approach especially in the era of emergence of Multidrug resistant organisms like MRSA, VRE, and carbapenem resistant GNB. *S.aureus* is one of the most commonly isolated bacteria from clinical infections which is associated with both acute and chronic infections and has the ability to form biofilms and develop drug resistance. MRSA isolates associated with biofilm formation are hard to eradicate, resulting in high morbidity and mortality. Compounds with both antibacterial and antibiofilm activity, which are less prone to develop resistance are the much-needed drugs for these infections.

In the present study, the new thiazolidine derivative TD-H2A has shown good bactericidal and anti-biofilm activity against various clinical and reference strains of *S.aureus* and other bacteria. The authors performed a series of experiments including MIC detection, time kill assays, and confocal laser scanning microscopy with live/dead cell staining for demonstrating the drug activity. Serial passaging experiments were conducted with increasing concentrations of the drug along with WGS sequencing to understand the kinetics of resistance development in *S.aureus*. The authors also evaluated the safety profile of the drug by investigating the cytotoxicity on Vero cells by using MTT assay and hemolytic properties on human RBC. Mouse skin infection model was used to evaluate the pathology by using H&E stain as well as detecting the viable bacterial counts. All these experiments being the strengths of the study, not knowing the precise mechanism of action of the compound is its major drawback.

Though It is a well conducted research study, the author might need to address the following minor issues

1. In the abstract (line 31) it is mentioned that a total of 40 non duplicate *S.aureus* were included in the study.. but the authors included other staphylococcal species as well as other bacteria from different genera for evaluation (Table1). It should be corrected.
2. Only Fig 1 legend is shown below the figure. Legends are missing for all the other figures.
3. Line 146 – mic should be capitalized
4. Line 151- figure 3c should be replaced by figure 2c as the membrane permeability at different time points is shown in fig 2c.
5. Line 169 – you find living CFU...should be replaced by appropriate words
6. Line 212- nasal cavity should be replaced by nasalcavity
7. Line 227- no little effect – should be either no effect or little effect
8. Line 241- mic₉₀ should be replaced by MIC₉₀ and valve should be replaced by value
9. Line 314- it is mentioned that no visible growth observed in the test tubes.. As per the methodology, MIC testing was done by broth microdilution method in 96 well plates... discrepancies should be corrected.
10. Line 396 &397 - It is preferable to keep the heading in bold letters.

Dear Reviewers,

Thank you very much for your time involved in reviewing the manuscript and your very encouraging comments on the merits. And thanks very much for your valuable advice too. I am appreciated. According to your comments, we have revised confusing languages from previous manuscript and inexact use of certain terms. We also appreciate your clear and detailed feedback and hope that the explanation has fully addressed all of your concerns. We hope our work can be improved again.

Furthermore, in the remainder of this letter, we discuss each of your comments individually along with our corresponding responses carefully and thoroughly.

To facilitate this discussion, we first retype your comments in italic font and then present our responses to the comments.

Reviewer #1 (Comments for the Author):

Dear Editor ,

Thank you for including me as a reviewer for the manuscript by Zhaoa et al., (Antimicrobial and Anti-biofilm Activity of A Thiazolidinone Derivative against Staphylococcus aureus). The manuscript discusses the development of the TD-H2-A derivative as a potential antimicrobial. The authors investigated the bactericidal and anti-biofilm activities of the thiazolidinone derivative (TD-H2-A) against S. aureus. 40 non-duplicate strains were collected, and the minimum inhibitory concentrations (MICs) of TD-H2-A were determined. The compound was found to have strong bactericidal activity against mature biofilms and low cytotoxicity and hemolytic activity against Vero cells and human erythrocytes. TD-H2-A also showed a good bactericidal effect on S. aureus SA113-infected mice. The authors concluded that TD-H2-A could be a potential agent to combat biofilm infections and multidrug-resistant staphylococcal infections.

Major comments:

1. Regarding the synthesized TD-H2-A, what was the purity of the compound obtained?

Response 1: Dear professor, thank you for your question. The purity of the synthetic derivative TD-H2-A was 95%.

2. Figure 2 A: Line 124: The survival rates of Vero cells after treatment with different concentrations ... Why the 12.5-25 Concentrations ($\mu\text{g}/\text{mL}$) had higher survival rates compared to 3.125-100 ($\mu\text{g}/\text{mL}$)? Are these differences statistically significant?

Response 2: Dear professor, thank you for your detailed review. We consider this result to be within the experimental margin of deviation. In order to be more rigorous, we repeated experiments for verification, and the relevant results and map are shown in FIG 2A.

3. Line 129: The hemolytic effect of the new synthetic derivatives was assessed on healthy human erythrocytes. "Concentration below the MIC did not have an obvious hemolytic effect on the hemolysis ratio of human red blood cells, which was less than 0.67%. At a higher concentration of 25µg /mL, the hemolysis ratio of the TD-H2-A derivative was less than 1.42%(data not shown)". Is there any reason to use up to 25µg/mL concentration for testing hemolysis effect and not up to 100 µg/mL like in the survival rate experiment of Vero cells in Figure 2-A? I feel the hemolysis data are important enough to be presented in a figure1.

Response 3: Dear reviewer, thank you for your wonderful advice. We selected 1×MIC and 4×MIC for hemolysis experiment according to the result of the resistance development assay. Thank you for your time, this is a very good suggestion. We have supplemented the experimental data in the supplementary FIG S1.

*4. Line 145: "By comparing the whole genome sequencing of HG001 treated with 4mic TD-H2-A on the 25th day and without treated TD-H2-A(Table S1), we found that two genes had a single-base mutation in *mnhD1*(Na⁺/H⁺ antiporter *Mnh1* subunit D) and a multi-base insertion mutation in *TrkA* family potassium uptake protein, while it was not reported that the correlation between the two genes and drug resistance". There is a lack of information about the genome sequencing method and the number of cultures sequenced. How many replicates were sequenced? There is a possibility of discovering more or other mutations if all the treatment cultures (replicates) were sequenced.*

Response 4: Dear professor, Thank you very much for the detailed review and your professional advice. There were three replicates sequenced in this experiment. The whole genome sequencing method and the number of cultures sequenced have been

replenished in the article. We repeated the experiment and sequenced the strains, and the newly discovered mutations have been completed in supplement Table S1.

507 **Whole genome sequencing.** We finished the whole genome sequencing of
508 HG001 treated with 4× MIC TD-H2-A on the 25th day and without treated TD-H2-A in
509 the resistance development. Sequencing was performed by Shanghai Biozeron
510 Biothchnology Co.Ltd(Shanghai, China). For Illumina pair-end sequencing of each
511 strain, at least 1µg genomic DNA was used for sequencing library construction. Paired-
512 end libraries with insert sizes of ~400bp were prepared following Illumina's standard
513 genomic DNA library preparation procedure. Purified genomic DNA is sheared into
514 smaller fragments with a desired size by Covaris, and blunt ends are generated by using
515 T4 DNA polymerase. After adding an 'A' base to the 3' end of the blunt phosphorylated
516 DNA fragments, adapters are ligated to the ends of the DNA fragments. The desired
517 fragments can be purified through gel-electrophoresis, then selectively enriched and
518 amplified by PCR. The index tag could be introduced into the adapter at the PCR stage
519 as appropriate and we did a library quality test. The sample was sequenced by the
520 Illumina NovaSeq 6000 platform (150bp*2, Shanghai Biozeron Biotechnology Co., Ltd,
521 Shanghai, China). There were three replicates sequenced in this experiment.

5. *Figure 3 A-F. The period of time points 8-24 hours is too long to be left without measuring OD.... What if there is growth improvement or decline during this unmeasured period of time? Because of that, the graph is squeezed and does not show the usual growth pattern of S. aureus in the control group. I think removing the points 8-24 or adding measurement points in between 8-24 hrs is helpful.*

Response 5: Dear professor, thank you for your advice. We referred to the following published literature for Time-killing curve determination(DOI: 10.1038/nature18634), therefore, we chose to measure at these time points. According to our experiment, bacterial growth has basically shown a downward trend at 8h. Thank you for your advice, and we will take it into consideration carefully in future experiments.

6. *Figure 3 F: Growth at 24 hours seems higher than at the 8 hrs point. Does it mean suppressors or mutations are starting to appear at this point?*

Response 6: Thank you for your detailed review. According to your question, we have done the experiment again and found that this was an experimental deviation, and we have made a modification.

7. Line 183 : *"The effects of the derivative on cell viability in mature S. aureus biofilms were detected using a CLSM with Live/Dead staining. In the 10× TD-H2-A treatment group, more than 99.4% of the bacteria in the biofilm were stained red by PI, indicating that the majority of the bacteria were dead (Fig. 3C)". How come the number of total cells ((Dead (red cells) and green (live cells)) in the TD-H2-A 1X MIC treatment is less than the total green cells in the control 1%DMSO or total red cells in the TD-H2-A 10 MIC treatment groups?*

Response 7: Dear professor, thank you for the detailed review. This phenomenon was caused by the capture difference in the field of view under the electron microscope. Sorry, we were not very thoughtful when choosing the field of view to photograph. We will take it into consideration carefully in future experiments.

8. Line 199: *H&E staining revealed that S. aureus SA113 induced infection in mouse skin tissues, whereas control mice receiving 1% DMSO had obviously positive staining. Needs further elaboration/Explanation.*

Response 8: Thank you for your time, this is a very good suggestion. We have added further explanations in the manuscript. DMSO is the solvent for TD-H2-A. Control mice receiving 1% DMSO had obviously positive staining, it indicates that 1% DMSO has no antibacterial effect.

Minor comments:

- *Language and typos: The English should be revised all throughout the manuscript for clarity and correct usage.*
- *Line 262. Because TD-H2-A had little effect on gram-negative bacteria. The statement is confusing .*
- *Line 171: The strains in the control group had been steadily growing (Fig.3A). I think this should be "The strains in the control group had been steadily growing (Fig.3A-F)"*
- *Figure 4-B Line 519 : Photos need numbering and referencing in the text.*

Response : Thank you so much for your great suggestions and detailed reviews, and

they have been corrected in the article.

Reviewer #2 (Comments for the Author):

The authors investigated the bactericidal and anti-biofilm activities of thiazolidinone derivative (TD-H2-A) against *S. aureus*. Overall, this is a clear, concise, manuscript. The introduction has sufficient information. The methods are generally appropriate, the results and discussion are clear and strongly supported by the data, here are my concerns:

1- I strongly recommend adding a graphical abstract to the manuscript.

Response 1: Thank you for your advice. Since the current mechanism is not very clear and the specific mechanism part is being studied, a roadmap will be added in future research.

2- For the title it would be better to add (in vitro and in vivo approach) to the end of it.

Response 2: Thanks very much for your great suggestion, and the title has been completed according to your advice.

3- Thiazolidinone-based molecules are attractive targets in the rational design of drug-like compounds, which possess anti-inflammatory, antioxidant, antitumor, choleric, diuretic, and other activities. In the current study, the authors repurposed them as antibiofilm and antibacterial agent, there is a lack of information about drug repurposing, the authors need to add short sentences to the introduction section about this approach as a new alternative approach to fight against antimicrobial resistance and the advantages of this approach over synthesis of new antibiotics. You can also support your text with these papers that use this approach;

**Kocking down *Pseudomonas aeruginosa* virulence by oral hypoglycemic metformin*

nano emulsion

**Alleviating the virulence of Pseudomonas aeruginosa and Staphylococcus aureus by ascorbic acid nanoemulsion*

Response 3: Thank you for your great suggestion. These opinions help to improve academic rigor and enrich the completeness of our article. We designed, synthesized, and screened the compound. And TD-H2-A is a new drug, not a re-use of an existing drug. Previously published literatures have been cited in the introduction to illustrate the origin of the compound.

4- It is better to mention from which patients the clinical isolates used in this study were obtained (with infection of the urogenital tract, diabetic foot, etc.).

Response 4: Thank you for your detailed review and great advice, and they have been complemented in this article. We have 10 clinical strains in total, including 2 sputum specimens, 2 whole blood specimens, 5 wound secretion specimens and 1 urine specimen. These strains isolated from these patients, including: preterm infants, type 2 diabetes mellitus, cerebral infarction, bronchial pneumonia, bar-gland abscess, fever, pneumonia, pyloric obstruction, dialysis preparatory medical treatment and dermatomyositis patients.

5- In mice model, 14 mice/group is really a huge number and unethically and 6-8 mice/group should be sufficient for statistics. In addition, please clarify in the methodology section why the treatment lasts only for 3 days. Was this being enough for the healing process?

Response 5: Thank you for your great suggestion. Initially, in order to reduce experimental deviation, we chose 14 mice in each group. In future animal experiments, we will fully consider your suggestion and redesign the number of experimental animals. Based on the time of drug onset in vitro, as well as the visual observation of mouse skin healing, we finally decided to treatment for 3 days.

6- It would be really great if the authors study the effect of TD-H2-A on the expression

level of genes controlling biofilm in *S. aureus* using qRT-PCR experiment. This will support your findings.

Response 6: Thank you for your great advice. According to your suggestion, we have carried out relevant experiments, and the results are shown in Figure 6. With 1×MIC TD-H2-A treated *S. aureus* (SA113), the expression levels of genes related to biofilm formation were decreased significantly.

491 **RNA extraction and reverse transcription-quantitative PCR.** *S. aureus* SA113
492 strains were cultured in TSB at 37°C for 12 h. The above overnight culture of *S. aureus*
493 SA113 strain was inoculated at 1:100 in 3 mL of TSB medium. After approximately 4
494 h of 37°C 200 rpm shaking incubation, TD-H2-A of 1× MIC were added for 15 min.
495 Positive control was the tubes where bacteria were inoculated with 1% DMSO. RNA
496 extraction was performed according to the manufacturer's instructions (EASYspin Plus
497 Bacterial RNA Rapid Extraction Kit, Proteinbio, Nanjing). Next, extracted RNA (1 µg
498 of each) was used as the template for cDNA synthesis by using a TRUEScript RT
499 MasterMix(OneStep gDNA Removal) (Proteinbio, Nanjing). Quantitative real-time
500 PCR (qRT-PCR) was performed by using the 2× Sybr Green (Proteinbio, Nanjing) and

Reviewer #3 (Comments for the Author):

In this study, the authors investigated the bactericidal and anti-biofilm activity of compound TD-H2-A, a derivative of thiazolidine, in various staphylococcus aureus strains. Repurposing of non-antibiotic group of drugs and their derivatives for the treatment of resistant bacterial infections has been a promising approach especially in the era of emergence of Multidrug resistant organisms like MRSA, VRE, and carbapenem resistant GNB. S.aureus is one of the most commonly isolated bacteria from clinical infections which is associated with both acute and chronic infections and has the ability to form biofilms and develop drug resistance. MRSA isolates associated with biofilm formation are hard to eradicate, resulting in high morbidity and mortality. Compounds with both antibacterial and antibiofilm activity, which are less prone to develop resistance are the much-needed drugs for these infections.

In the present study, the new thiazolidine derivative TD-H2A has shown good bactericidal and anti-biofilm activity against various clinical and reference strains of

S.aureus and other bacteria. The authors performed a series of experiments including MIC detection, time kill assays, and confocal laser scanning microscopy with live/dead cell staining for demonstrating the drug activity. Serial passaging experiments were conducted with increasing concentrations of the drug along with WGS sequencing to understand the kinetics of resistance development in S.aureus. The authors also evaluated the safety profile of the drug by investigating the cytotoxicity on Vero cells by using MTT assay and hemolytic properties on human RBC. Mouse skin infection model was used to evaluate the pathology by using H&E stain as well as detecting the viable bacterial counts. All these experiments being the strengths of the study, not knowing the precise mechanism of action of the compound is its major drawback.

Though It is a well conducted research study, the author might need to address the following minor issues.

1. In the abstract (line 31) it is mentioned that a total of 40 non duplicate S.aureus were included in the study. but the authors included other staphylococcal species as well as other bacteria from different genera for evaluation (Table1). It should be corrected.

Response 1: Thank you for your detailed review, and it has been modified in the manuscript.

34 A total of 40 non-duplicate ~~S.aureus~~ strains were collected, and the minimum inhibitory
35 concentrations (MICs) of TD-H2-A were determined. The effect of TD-H2-A on

2. Only Fig 1 legend is shown below the figure. Legends are missing for all the other figures.

Response 2: Thank you for your detailed review. All the legends are in the Table and figures text of the manuscript, and we have removed the Figure 1 legend.

3. Line 146 - mic should be capitalized.

4. Line 151- figure 3c should be replaced by figure 2c as the membrane permeability at different time points is shown in fig 2c.
5. Line 169 - you find living CFU...should be replaced by appropriate words.
6. Line 212- nasal cavity should be replaced by nasal cavity.
7. Line 227- no little effect - should be either no effect or little effect.
8. Line 241- mic90 should be replaced by MIC90 and value should be replaced by value.
9. Line 314- it is mentioned that no visible growth observed in the test tubes.. As per the methodology, MIC testing was done by broth microdilution method in 96 well plates... discrepancies should be corrected.
10. Line 396 &397 - It is preferable to keep the heading in bold letters.

Response 3-10: Thank you for the detailed reviews, and they have been corrected in the article.

Reviewer #4 (Comments for the Author):

The manuscript by Zhao et al. continues from previous work in the development of an antimicrobial compound that can be effective against Staphylococcus aureus within a biofilm. The authors demonstrate that their new derivative, TD-H2-A, has both bactericidal and anti-biofilm activity.

Recommendations:

1. *The authors should explain in the manuscript why S. aureus HG001 was used in some experiments while S. aureus SA113 was used in other experiments.*

Response 1: Thank you for your detailed review and suggestion. *S. aureus* SA113 has a strong ability of biofilm formation. So we conducted a series of biofilm-related experiments with *S. aureus* SA113. We referred to the following published literature for resistance test which used *S. aureus* HG001 to determine whether TD-H2-A resistant colonies can be generated (DOI: 10.3389/fmicb.2019.00014).

2. *The term MIC90 is misused in the manuscript. MIC90 refers to the concentration of a compound which inhibits 90% of the tested bacterial strains, and it is best practice that data from at least 10 strains of a single species are used to calculate the MIC90. The term MIC should be used instead on lines 39, 112, 241, and in Table 1. As written, the results section does not flow particularly well. The primary reason for this is that the individual sub-sections feel disjointed from one another. This could be remedied with the addition of sentences at the beginning and end of each sub-section, which would serve to provide context and justification for performing the experiments and also serve as bridges between the sub-sections.*

Response 2: Dear professor, thank you for your advice. According to your comments, we have revised inexact use of certain terms from previous literature.

3. *Line 31: Table 1 indicates that the strain collection consisted of 20 S. aureus strains and 20 strains encompassing many other genera and species, not 40 non-duplicate S. aureus strains, as stated on line 31.*

Response 3: Thank you for your detailed review, and it has been modified in the manuscript.

4. *Abstract: The target of TD-H2-A should be stated in the abstract.*

Response 4: Thank you very much for your valuable advice. TD-H2-A is drug obtained through high-throughput screening, but we have not yet defined the target of the drug, the target of TD-H2-A is probably the WalK/WalR TCS, so it is not reflected in the abstract.

5. *Lines 74-75: It is not accurate that the biofilm matrix restricts penetration of host defense molecules and antimicrobial agents. Many molecules are able to penetrate the full depth of biofilms but are inactive due to bacterial cell metabolic state or other reasons.*

Response 5: Thank you very much for your detailed review. We have removed the inappropriate part.

6. Paragraph beginning line 110: The authors should define what they consider to be "potent antimicrobial activity" (line 114-115) of the TD-H2-A compound.

Response 6: Dear professor, thank you very much for your detailed review. The TD-H2-A has an obvious effect on *Streptococcus pneumoniae*, but isn't particular good effect on gram-negative bacteria as seen from the Table 1. We have revised it in the text.

7. Lines 145-150 are difficult to understand as written and could benefit from revision. In addition, the authors should clarify: (i) What does 4mic mean? Do the authors mean to say 4x MIC?, (ii) What does "without treated TD-H2-A" mean? Was this the HG001 isolate from Day 0?, (iii) Line 147 says two genes had a single base mutation in the *mnhD1* gene, which does not make sense and is not consistent with data in the supplemental table., (iv) The insertion mutation was in a gene, not a protein (line 149).

Response 7: Dear professor, thank you very much for your detailed review and advice. (i)The 4mic have been replaced by 4× MIC. (ii)Yes, this is mean the HG001 isolate from Day 0(which is no treated with TD-H2-A). (iii)We gained six mutations, including:

BSR30_RS01405,rrf,mnhD1,BSR30_RS05085,BSR30_RS07385,BSR30_RS12160, and it was not reported that the correlation between the six genes and drug resistance. These mutated genes did not seem to be associated with bacterial drug resistance, and the sequencing comparison results are shown in supplemental Table S1. (iv) We have corrected it in the article.

8. Line 151: Should this be Figure 2C, not Figure 3C?

Response 8:Thank you for your detailed review, and it has been modified in the manuscript.

9. Lines 173-180: The *p* values on lines 175 and 178 do not make sense with the comparisons as written in this section. Is it possible the *p* values are misplaced?

Response 9:Thank you for your detailed review, and it has been modified in the

manuscript.

10. Line 191: *This is a subjective assessment and should be removed or replaced with a quantitative assessment.*

Response 10: Thank you for your great suggestion, and it has been removed in the manuscript.

11. Line 194: *A few sentences explaining the mouse infection model is needed at the start of the "Mouse infection model and pathological sections" sub-section.*

Response 11: Thank you for your great advice, and we have added a few sentences to flesh out our article.

199 **Mouse infection model and pathological sections.** To identify the further efficacy
200 of TD-H2-A in the skin and soft tissue infection murine model, we performed a
201 experiment in which the mice were subjected to TD-H2-A regimens. And we also
202 performed hematoxylin and eosin (H&E) staining for histological examination
203 assessing wound healing. Comparing the treatment group to the 1% DMSO control
204 group, there was no significant difference between the vancomycin and control groups
205 ($p > 0.05$), while the number of SA113 viable bacteria in the skin of mice of the TD-
206 H2-A group was significantly reduced which had a fourfold reduction ($p = 0.0005$; Fig.

12. *The sentence on lines 199-200 does not make sense as written. Are the control mice that received 1% DMSO a control for the infection (i.e., received no S. aureus) or for the treatment (i.e., were infected with S. aureus but received only vehicle)?*

Response 12: Dear professor, thank you very much for your detailed review. The control mice were infected with *S. aureus* but received only vehicle (1% DMSO).

13. *Sentence on lines 221-222: Reference 24 is not an appropriate citation for this statement.*

Response 13: Thank you for your great suggestion, and reference 24 has been removed in the manuscript.

14. *Paragraph on lines 237-244: This paragraph is confusing as written, as sentences about the activity of TD-H2-A against S. aureus biofilms are interrupted with statements about the MIC of the compound when tested against other genera. On line*

244, please specify which bacterial species it was in which 99.4% of the biofilm bacteria were killed.

Response 14: Thank you for your great suggestion, we have removed the reference to *Streptococcus pneumoniae* from that paragraph. and *S. aureus* (SA113) bacterial specie has been specified in the article.

15. Lines 261-263: The authors did not show that increased cell permeability increased drug absorption, and it is unclear how the following statement about the TD-H2-A having little effect on Gram-negative bacteria is related to cell permeability.

Response 15: Dear professor, thank you very much for your detailed review. We have make a correction.

277 cell membrane permeability at 4× and 8× MICs. ~~The cell membrane permeability is~~
278 ~~increased which increased drug absorption, but it seem not to be the main antibacterial~~
279 ~~mechanism. Because TD-H2-A had little effect on gram negative bacteria.~~

16. Lines 264-269: The information in this paragraph contradicts lines 138-150, Figure 2B, and the supplemental table, where it is indicated that the *S. aureus* strain passaged in TD-H2-A showed reduced susceptibility and gained two mutations.

Response 16: Dear professor, thank you very much for your detailed review. According to the sequencing results, mutations have occurred in some genes, including:

BSR30_RS01405, rrf, mnhD1, BSR30_RS05085, BSR30_RS07385, BSR30_RS12160, and it was not reported that the correlation between the six genes and drug resistance.

17. Line 273: The authors should be more specific than saying "According to the aforementioned methods".

Response 17: Thank you for your great suggestion, and it has been specified in the article.

292 suggesting the HAMP domain. We hypothesize that it interacts with the HAMP domain to
293 exert its bactericidal effect. According to the aforementioned methods, We constructed
294 the WalK protein dimer model and the WalK protein-drug small molecule complex
295 molecular docking model. The WalK protein dimer structure model was generated
296 using the protein complex structure prediction tool AlphaFold-multimer (40). To screen
297 for drug binding sites, possible pockets of the dimer model were calculated by mDPA
298 (41). To further investigate the role of small drug molecules in the ATP binding pocket
299 of the CA domain, Modeler was used to model the monomeric CA domain (42), and
300 AutoDock Vina was used for docking (43). TD-H2-A binds to the CA domain, as well
301 as the HAMP domain. But we yet don't know the exact mechanism is still unknown.

18. Line 292: The authors should provide details on how the TD-H2-A compound was obtained for use in this study.

Response 18: Thank you for your great advice. Previously published literatures have been cited in the introduction to illustrate the origin of the compound.

19. The sentence on L293-294 needs to be revised, "The derivatives' structures and systematic names are depicted in Figure 1." This sentence appears to have been pulled from a previous study where multiple derivatives were made and examined, whereas here only a single compound is studied.

Response 19: Thank you for your detailed review, and it has been modified in the manuscript.

20. Line 316: In what way was *S. aureus* ATCC25923 used for quality control in the MIC assays? Was it tested with a particular antibiotic? Quality control cannot be assured if a known drug/agent is not used.

Response 20: Thank you for your detailed review, *S. aureus* ATCC25923 was determined by broth dilution method in the MIC assays for quality control, and it tested Vancomycin as a particular antibiotic.

21. Line 362: Details of whole genome sequencing methods are needed. The genome sequence results need to be deposited in a repository for public access.

Response 21: Thank you for your great suggestion, the information about the genome

sequencing method has been supplemented in the text.

507 **Whole genome sequencing.** We finished the whole genome sequencing of
508 HG001 treated with 4× MIC TD-H2-A on the 25th day and without treated TD-H2-A in
509 the resistance development. Sequencing was performed by Shanghai Biozeron
510 Biothchnology Co.Ltd(Shanghai, China). For Illumina pair-end sequencing of each
511 strain, at least 1µg genomic DNA was used for sequencing library construction. Paired-
512 end libraries with insert sizes of ~400bp were prepared following Illumina's standard
513 genomic DNA library preparation procedure. Purified genomic DNA is sheared into
514 smaller fragments with a desired size by Covaris, and blunt ends are generated by using
515 T4 DNA polymerase. After adding an 'A' base to the 3' end of the blunt phosphorylated
516 DNA fragments, adapters are ligated to the ends of the DNA fragments. The desired
517 fragments can be purified through gel-electrophoresis, then selectively enriched and
518 amplified by PCR. The index tag could be introduced into the adapter at the PCR stage
519 as appropriate and we did a library quality test. The sample was sequenced by the
520 Illumina NovaSeq 6000 platform (150bp*2, Shanghai Biozeron Biotechnology Co., Ltd,
521 Shanghai, China). There were three replicates sequenced in this experiment.

22. *Lines 396-397: Should these lines be formatted as a section header?*

Response 22: Thank you for your great suggestion, and these spaces have been removed in the manuscript.

23. *In L405 "ultrasonically" should be changed to sonicated. Line 405: What volume and liquid were used for the sonication step? Was sonication performed with a probe or bath sonicator?*

Response 23: Thank you for your detailed review, and it has been supplemented in the article. The volume is 1mL per sample.

24. *Line 470, Statistical analysis paragraph: One or more figure legends indicate that an ANOVA test was used, but it is not mentioned in this paragraph.*

Response 24: Thank you for your detailed review, and it has been supplemented in the article.

25. *The Figure 2 and Figure 3 legends each contain several lines of text summarizing the results. This text should be moved to the results and deleted from the legends.*

Response 25: Thank you for your suggestion, we have made adjustments in the text.

26. *Figures 2 and 4 legends: Please state the statistical test used in each figure.*

Response 26: Thank you for your detailed review, and they have been supplemented in the article.

27. Line 506: *There is no blue line in the graph. The vancomycin treatment data are shown as red or pink.*

28. L507 *"** indicates $p < 0.01$, *** $p < 0.001$," ** is not used in the referenced figure (Fig. 3B), and should be removed from the figure description. Additionally, an explanation of what **** indicates should be added here as it is utilized in the figure, but not explained in the figure description.*

29. On L518 *"** $p < 0.01$ " should be removed as it is not used in the referenced figure (Fig. 4A). Additionally, an explanation should be added to this figure legend to explain the meaning of "NS".*

Response 27-29: Thank you for your detailed reviews, and they have been modified in the manuscript.

30. *In Fig. 2B the results of only a single experiment are shown. If possible, the results of all three replicates should be combined and shown here.*

Response 30: Dear professor, thank you very much for your detailed review. We repeated the resistance development experiment during the revision period. The graph shown in the figure is a single experiment that represents three replicates.

31. *In Figs. 2C and 3A the concentrations are listed as "5*MIC", whereas within the text and Fig. 3C the concentrations are listed as "5x MIC". The labeling in Fig. 3 should be changed to be in line with the rest of the manuscript.*

32. *There is a discrepancy between the labeling of Figs. 2C, 3B, and 4A. Figs. 2C and 3B use "ns" to indicate a p -value > 0.05 , while Fig. 4A uses "NS". This labeling should be made consistent across figures.*

33. *Table 1: Why is the MIC listed as 25.0 for some strains but 25.04 for other strains? In addition, why is MBC data given in the table footnote? No methods describing MBC have been included in the manuscript.*

34. *Supplementary table: Several columns are empty and should be removed. Give the definitions of the abbreviations used in the headers for the columns that do contain data. In the table title, does '4mic' mean '4xMIC'?_*

35. *Figure 2A x-axis should be labeled as "Agent Concentration". Figure 2C y-axis should be labeled as "Green/Red Ratio".*

36. *The labeling of the six graphs as lowercase a-f in Figure 3A is confusing.*

37. *The order of strains should be the same in Figures 4A and 4B, i.e., 1% DMSO on the left, vancomycin in the middle, TD-H2-A on the right.*

38. *Significant attention should be given to the writing of the manuscript, as there are currently multiple typos and grammatical errors throughout. In addition, the following specific revisions are needed for improved clarity or accuracy:*

a) Lines 56-60 is a long sentence.

b) Line 72 need to add word to read "S. aureus infections."

c) Lines 124-129 contain redundant information.

d) Line 169 revise remove informal writing "you find living"

e) Line 174 revise to remove informal writing "can hardly be"

f) Line 191 revise to remove informal writing "mostly died" and "basically died"

g) Lines 212-216 should be broken into shorter sentences.

h) Line 212 nasal, not nosal

i) Line 217 need to revise to read "that comprises the biofilm matrix"

j) Line 225: Replace Clostridium with Clostridioides.

k) The first sentence in the paragraph on line 253 is unnecessary.

l) Line 276 revise to remove informal writing "But we yet don't know"

m) Line 297: VRE should be replaced with the bacterial genus and species.

n) Line 331: Revise to say "cells treated with nisin..."

Response31-38: Dear professor, thank you very much for your detailed reviews and great suggestions on improving the accessibility of our manuscript. These opinions help to improve academic rigor and enrich the completeness of our article. Minor errors and irregularities have been corrected by us carefully and thoroughly in the article.

59 *Staphylococcus aureus* (*S. aureus*) is a major commensal bacterium and a human
60 pathogen that causes a hard-to-estimate number of uncomplicated skin infections and
61 hundreds of thousands to millions of severe invasive infections per year (1-3), including
62 pneumonia and other respiratory tract infections, surgical site, prosthetic joint, and
63 cardiovascular infections, as well as nosocomial bacteremia. Antibiotic resistance is a

Thank you very much for the detailed review and your professional advice. Your great suggestion and shared references are very useful. We would like to take this opportunity to thank you for all your time involved and this great opportunity for us to improve the manuscript. Based on your suggestion and request, we have made corrected modifications on the revised manuscript. We hope you will find this revised version satisfactory.

Sincerely,

The Authors

November 5, 2023

Re: Spectrum02327-23R1 (Antimicrobial and Anti-biofilm Activity of A Thiazolidinone Derivative against *Staphylococcus aureus* *in vitro* and *in vivo*)

Dear Dr. Yanfeng Zhao:

Thank you for the privilege of reviewing your work. Below you will find my comments, instructions from the Spectrum editorial office, and the reviewer comments.

Revision Guidelines

Sincerely,
Tomefa Asempa
Editor
Microbiology Spectrum

Thank you for the modifications. Paper is almost acceptable. 3 reviewers found your revisions to be acceptable however I would recommend that the following revisions be made prior to consideration for acceptance:

Reviewer #4 (Comments for the Author):

The authors have made many of the requested revisions to the manuscript. The key points made within the manuscript are now clearer and, overall, the manuscript is significantly improved. However, the manuscript could be further improved with additional edits, as listed below.

In the Methods section beginning line 299, the source and purity of the TD-H2-A used in this study must be stated here.

Figure 5B: An H&E section from an uninfected mouse would be a useful addition to consider to this data set.

Table S1: All abbreviations used in the table (including the row and column headers) need to be defined as footnotes to the table.

In each instance where the animal model used is referred to (the Abstract line 40, Introduction line 108, Discussion line 278, and Methods line 442), the language used makes it sound like the model used was newly developed and optimized specifically for this work. Is this the case? If not, and this model is based on one used previously, then the previous work should be cited.

Line 44, where it states "TD-H2-A killing almost all planktonic *S. aureus* USA300 at 5x MIC", needs revision. It is unclear to this reviewer what this sentence means.

Line 105 describes some of the bacterial strains studies as "standard strains". What does "standard" mean? Often the term "laboratory strain" or the descriptor "well characterized strain" are used in the literature when referring to strains that have been used in prior studies or have been used in laboratories for many years.

The paragraph in the results describing the mouse experiments, lines 195-203, should be edited throughout for clarity.

The sentence on lines 274-275 needs to be revised for clarity. Is it meant to express that the six genes identified had no previously known correlation to drug resistance, or that the authors found no correlation between these genes and drug resistance?

Methods, paragraph beginning line 411: This methods section needs some clarifications. Firstly, on line 413, it only states that the biofilm was formed for 24 hours. Enough detail to ensure independent replication by an independent lab needs to be provided. Secondly, lines 414-416 describe an assay in which the derivative was mixed with the bacterial cells in broth culture, then those were added to multi-well dishes to form biofilms for 16 hours. However, this section is supposed to describe the methods for testing of the derivative against a mature biofilm. Mixing the derivative with cells in broth culture then inoculating into a polystyrene well to form a biofilm is not testing the derivative against a mature biofilm.

The sentence in the Methods on lines 501-503 needs to be revised for clarity.

Figure 4 figure legend, lines 552-555: These sentences describe results and should be removed from the figure legend, as this information is already found in the text.

Figure 5 legend, line 563: A student's t-test is an inappropriate statistical test to compare more than 2 groups. This figure compared 5 conditions, which requires an ANOVA with appropriate posthoc test. The statistical analysis of these results needs to be redone, with the revised outcome accurately reported in the figure legend and updated on the figure, if applicable.

Figure 6 legend, line 566: The statistical test used to analyze these data to obtain the indicated p-value needs to be stated. It would also be helpful if the number of independent biological replicates performed and what the error bars represent are also stated.

The following edits would also improve the clarity of the paper:

- a. Line 40: "Animal models were constructed" should be changed to "A mouse skin infection model was used"
- b. Line 54: Delete the word "to" that is prior to "develop drug resistance".
- c. Line 122: Replace the words "those of" with "the activities against".
- d. Lines 136-137: Revise to "caused minimal cytotoxicity to Vero cells and minimal hemolytic activity against human erythrocytes"
- e. Lines 143-145 the word "is" should be replaced with "indicates"
- f. On line 150 "no" should be changed to "not"
- g. Line 151: Please state the exact number of genes instead of saying "some genes".
- h. On line 197 "mices" should be changed to "mice"
- i. On line 205 "mouse" should be changed to "mice"
- j. On line 263 "It" should be lower case, and "less hemolytic activity for" should be replaced with "minimal hemolytic activity on".
- k. On line 276 "producting" should be changed to "producing"
- l. Lines 277-278: "by constructing" should be changed to "using"
- m. On line 346 "solvent" should be removed
- n. On line 382-383 "stains all cells green" and "stains cells with damaged membranes red" can be removed as they are unnecessary
- o. On line 390-391 "an amphiphilic peptide with strong antimicrobial activity against various Gram-positive bacteria" is unnecessary and can be removed.
- p. Line 442: Revise to read "A mouse model of skin and soft tissue infection induced by *S. aureus* SA113 subcutaneous inoculation was used."
- q. Line 496: Replace "endogenous" with "constitutively expressed" or "housekeeping". All of the genes tested for expression were 'endogenous,' in that they are part of the *S. aureus* genome.

Reviewer comments

All the issues raised by the reviewers are adequately addressed in the revised manuscript

Dear Reviewer,

Thank you very much for your time involved in reviewing the manuscript and your very encouraging comments on the merits. And thanks very much for your valuable advice too. According to your comments, we have revised confusing languages from previous manuscript and inexact use of certain terms. We also appreciate your clear and detailed feedback and hope that the explanation has fully addressed all of your concerns. Furthermore, in the remainder of this letter, we discuss each of your comments individually along with our corresponding responses carefully and thoroughly. To facilitate this discussion, we first retype your comments in italic font and then present our responses to the comments.

Reviewer #4 (Comments for the Author):

The authors have made many of the requested revisions to the manuscript. The key points made within the manuscript are now clearer and, overall, the manuscript is significantly improved. However, the manuscript could be further improved with additional edits, as listed below.

In the Methods section beginning line 299, the source and purity of the TD-H2-A used in this study must be stated here.

Response: Dear professor, thank you for your detailed review, and we have put the source and purity of the TD-H2-A in the supplementary materials.

Figure 5B: An H&E section from an uninfected mouse would be a useful addition to consider to this data set.

Response: Thank you for your advice, and we have added an H&E section from an uninfected mouse to this data set.

Table S1: All abbreviations used in the table (including the row and column headers) need to be defined as footnotes to the table.

Response: Thank you for your advice, and we have added footnotes for the abbreviations in table S1.

In each instance where the animal model used is referred to (the Abstract line 40, Introduction line 108, Discussion line 278, and Methods line 442), the language used makes it sound like the model used was newly developed and optimized specifically for this work. Is this the case? If not, and this model is based on one used previously, then the previous work should be cited.

Response: Dear professor, thank you for your detailed review. Yes, this was an optimized animal model based on some previously reported experiments. Previously published literatures have been cited in the Methods.

Line 44, where it states "TD-H2-A killing almost all planktonic S. aureus USA300 at 5 ×MIC", needs revision. It is unclear to this reviewer what this sentence means.

Response: Dear professor, Thank you very much for the detailed review and your professional advice. It means “The 5×MIC TD-H2-A killed almost all planktonic *S. aureus* USA300”, and it have been corrected in the article.

45 The MIC values of TD-H2-A against the different *S. aureus* strains were 6.3–25.0
46 µg/mL. The 5× MIC TD-H2-A killed almost all planktonic *S. aureus* USA300. The

Line 105 describes some of the bacterial strains studies as "standard strains". What does "standard" mean? Often the term "laboratory strain" or the descriptor "well characterized strain" are used in the literature when referring to strains that have been used in prior studies or have been used in laboratories for many years.

Response: Thank you for your detailed review and professional advice. The standard strains refers to laboratory strain, and it has been modified in the manuscript.

The paragraph in the results describing the mouse experiments, lines 195-203, should be edited throughout for clarity.

Response: Dear professor, thank you for your detailed review, and they have been modified in the manuscript.

198 **Mouse infection model and pathological sections.** To evaluate the effect of
199 TD-H2-A in vivo, we used a mouse skin infection model in which the mice were
200 injected with TD-H2-A, vancomycin, and 1% DMSO respectively. And we also
201 performed hematoxylin and eosin (H&E) staining for assessing treatment effect.
202 Comparing each treatment group (vancomycin group and TD-H2-A group) to the 1%
203 DMSO control group, there was no significant difference between the vancomycin
204 and control groups ($p > 0.05$), while the number of SA113 viable bacteria in the skin
205 of mice of the TD-H2-A group was significantly reduced which had a fourfold
206 reduction ($p = 0.0005$; Fig. 5A).

The sentence on lines 274-275 needs to be revised for clarity. Is it meant to express that the six genes identified had no previously known correlation to drug resistance, or that the authors found no correlation between these genes and drug resistance?

Response: Dear professor, thank you for your detailed review. It means “The six genes identified had no previously known correlation to drug resistance”, and it have been corrected in the article.

277 sequencing. According to the sequencing results, there were six genes that were
278 mutated and they identified had no previously known correlation to drug
279 resistance. TD-H2-A has potent activity against multiple resistant Gram positive

Methods, paragraph beginning line 411: This methods section needs some clarifications. Firstly, on line 413, it only states that the biofilm was formed for 24 hours. Enough detail to ensure independent replication by an independent lab needs to be provided. Secondly, lines 414-416 describe an assay in which the derivative was mixed with the bacterial cells in broth culture, then those were added to multi-well

dishes to form biofilms for 16 hours. However, this section is supposed to describe the methods for testing of the derivative against a mature biofilm. Mixing the derivative with cells in broth culture then inoculating into a polystyrene well to form a biofilm is not testing the derivative against a mature biofilm.

Response: Thank you for your detailed review and your professional advice. We performed the biofilm experiments using the same method described in the published paper (DOI: 10.1038/emi.2015.1). And we validated the biofilm formed at 24h by crystal violet staining in this experiment.

The bacterial cultures in TSB medium had formed the biofilm for 24h. Then mixing the derivative with cells in broth culture and inoculating into a polystyrene well for 16 hours.

415 glucose-containing TSB medium. Then the biofilm was formed for 24h(18). Aliquots
416 of the inhibitors at 1×, 5×, and 10× MIC concentrations were mixed with the same

417 volume of the bacterial cultures in TSB medium, added in triplicate to a 12-well
418 polystyrene plate (1mL /well), and statically incubated for 16 hours. As a positive

The sentence in the Methods on lines 501-503 needs to be revised for clarity.

Response: Thank you for your detailed review and professional advice, and it has been modified in the manuscript.

503 **Whole genome sequencing.** We analyzed the whole genome sequencing of
504 HG001 treated with 4× MIC TD-H2-A on the 25th day and not treated with
505 TD-H2-A. Sequencing was performed by Shanghai Biozeron Biothchnology

Figure 4 figure legend, lines 552-555: These sentences describe results and should be removed from the figure legend, as this information is already found in the text.

Response: Thank you very much for your advice. We have removed them in the manuscript.

Figure 5 legend, line 563: A student's t-test is an inappropriate statistical test to compare more than 2 groups. This figure compared 5 conditions, which requires an ANOVA with appropriate posthoc test. The statistical analysis of these results needs to be redone, with the revised outcome reported accurately reported in the figure legend and updated on the figure, if applicable.

Response: Dear professor, thank you for your detailed review. We compared each treatment group (vancomycin group and TD-H2-A group) with 1%DMSO control group for statistical analysis, and there were a comparison between the two groups.

Figure 6 legend, line 566: The statistical test used to analyze these data to obtain the indicated *p*-value needs to be stated. It would also be helpful if the number of independent biological replicates performed and what the error bars represent are also stated.

Response: Thank you very much for your advice. The statistical test used to analyze these data and what the error bars represent are have been completed in the article.

565 **Figure 6.** qRT-PCR results of TD-H2-A's effect on the transcription of biofilm-related
566 genes. All assays in were performed with three biologically independent experiments,
567 and the mean \pm SD is shown. Statistical differences between control and antibiotic
568 treatment groups were analyzed by one-way ANOVA with Tukey's multiple
569 comparisons test (***P* < 0.001).

The following edits would also improve the clarity of the paper:

- a. Line 40: "Animal models were constructed" should be changed to "A mouse skin infection model was used"
- b. Line 54: Delete the word "to" that is prior to "develop drug resistance".
- c. Line 122: Replace the words "those of" with "the activities against".
- d. Lines 136-137: Revise to "caused minimal cytotoxicity to Vero cells and minimal hemolytic activity against human erythrocytes"
- e. Lines 143-145 the word "is" should be replaced with "indicates"
- f. On line 150 "no" should be changed to "not"
- g. Line 151: Please state the exact number of genes instead of saying "some genes".
- h. On line 197 "mices" should be changed to "mice"
- i. On line 205 "mouse" should be changed to "mice"
- j. On line 263 "It" should be lower case, and "less hemolytic activity for" should be replaced with "minimal hemolytic activity on".
- k. On line 276 "producing" should be changed to "producing"
- l. Lines 277-278: "by constructing" should be changed to "using"
- m. On line 346 "solvent" should be removed
- n. On line 382-383 "stains all cells green" and "stains cells with damaged membranes red" can be removed as they are unnecessary
- o. On line 390-391 "an amphiphilic peptide with strong antimicrobial activity against various Gram-positive bacteria" is unnecessary and can be removed.
- p. Line 442: Revise to read "A mouse model of skin and soft tissue infection induced by *S. aureus* SA113 subcutaneous inoculation was used."
- q. Line 496: Replace "endogenous" with "constitutively expressed" or "housekeeping". All of the genes tested for expression were 'endogenous,' in that they are part of the *S. aureus* genome.

Response: Dear professor, thank you very much for your detailed reviews and great suggestions on improving the accessibility of our manuscript. These opinions help to improve the clarity of the paper and enrich the completeness of our article. Minor errors and irregularities have been corrected by us carefully and thoroughly in the article.

Re: Spectrum02327-23R2 (Antimicrobial and Anti-biofilm Activity of A Thiazolidinone Derivative against *Staphylococcus aureus* *in vitro* and *in vivo*)

Dear Dr. Yanfeng Zhao:

Your manuscript has been accepted, and I am forwarding it to the ASM production staff for publication. Your paper will first be checked to make sure all elements meet the technical requirements. ASM staff will contact you if anything needs to be revised before copyediting and production can begin. Otherwise, you will be notified when your proofs are ready to be viewed.

Sincerely,
Tomefa Asempa
Editor
Microbiology Spectrum